# Bayesian Optimization with High-Dimensional Outputs

**Wesley J. Maddox**
New York University
wjm363@nyu.edu

**Maximilian Balandat**
Facebook
balandat@fb.com

**Andrew Gordon Wilson**
New York University
andrewgw@cims.nyu.edu

**Eytan Bakshy**
Facebook
eytan@fb.com

## Abstract

Bayesian Optimization is a sample-efficient black-box optimization procedure that is typically applied to problems with a small number of independent objectives. However, in practice we often wish to optimize objectives defined over many correlated outcomes (or "tasks"). For example, network operators may want to optimize the coverage of a cell tower network across a dense grid of locations. Similarly, engineers may seek to balance the performance of a robot across dozens of different environments via constrained or robust optimization. However, the Gaussian Process (GP) models typically used as probabilistic surrogates for multi-task Bayesian Optimization scale poorly with the number of outcomes, which greatly limitis their applicability. We devise an efficient technique for exact multi-task GP sampling that combines exploiting Kronecker structure in the covariance matrices with Matheron's identity, allowing us to perform Bayesian Optimization using exact multi-task GP models with tens of thousands of correlated outputs. In doing so, we achieve substantial improvements in sample efficiency compared to existing approaches that only model aggregate functions of the outcomes. We demonstrate how this unlocks a new class of applications for Bayesian Optimization across a range of tasks in science and engineering, including optimizing interference patterns of an optical interferometer with more than 65,000 outputs.

## 1 Introduction

Many problems in science and engineering involve reasoning about multiple, correlated outputs. For example, cell towers broadcast signal across an area, and thus signal strength is spatially correlated. In randomized experiments, treatment effects on multiple outcomes are naturally correlated due to shared causal mechanisms. Without further knowledge of the internal mechanisms (i.e., in a "black-box" setting), Multi-task Gaussian processes (MTGPs) are a natural model for these types of problems as they model the relationship between each output (or "task") while maintaining the gold standard predictive capability and uncertainty quantification of Gaussian processes (GPs). Many downstream analyses require more of the model than just prediction; they also involve sampling from the posterior distribution to estimate quantities of interest. For instance, we may be interested in the performance of a complex stock trading strategy that requires modeling different stock prices jointly, and want to characterize its conditional value at risk (CVaR) [49], which generally requires Monte Carlo (MC) estimation strategies [10]. Or, we want to use MTGPs in Bayesian Optimization (BO), a method for sample-efficient optimization of black-box functions. Many state of the art BO approaches use MC acquisition functions [60, 3, 4], which require sampling from the posterior distribution over new candidate data points.

Drawing posterior samples from MTGPs means sampling over all tasks and all new data points, which typically scales *multiplicatively* in the number of tasks ($t$) and test data points ($n$), e.g. like $\mathcal{O}(n^3t^3)$ [52, 6]. For problems with more than a few tasks, posterior sampling thus quickly becomes

intractable due to the size of the posterior covariance matrix. This is especially problematic in the case of many real-world problems that can have hundreds or thousands of correlated outputs that should be modelled jointly in order to achieve the best performance.

For instance, the cell tower signal maps in Figure 1 each contain 2,500 outputs (pixels). In this problem, we aim to jointly tune the down-tilt angle and transmission power of the antennas on each cell tower (locations shown in red) to optimize a global coverage quality metric, which is a known function of power and interference at each location [21]. Since simulating the power and interference maps given a parameterization is computationally costly, traditionally one might apply BO to optimize the aggregate metric. At its core, this problem is a composite BO problem [3, 4], so we expect an approach that models the constituent outcomes at each pixel individually to achieve higher sample efficiency. However, modelling each pixel using existing approaches used for BO is completely intractable in this setting, as we would have to train and sample from a MTGP with 5,000 tasks.

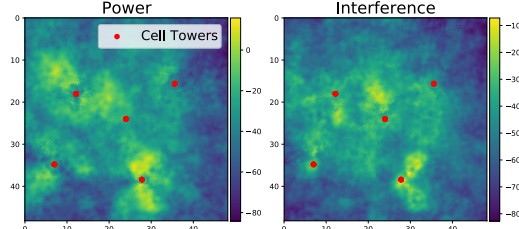

Figure 1: Map of radio signal power and interference for fixed locations of cell towers (red dots). Outcomes vary smoothly with respect to the towers' down-tilt angle and transmission power. Our goal is to optimize statistics of these maps as to maximize the overall signal coverage across an area while minimizing interference.

To remedy the poor computational scaling with the number of tasks, we exploit Matheron's rule for sampling from GP posterior distributions [13, 59]. We derive an efficient method for MTGP sampling that exploits Kronecker structure inherent to the posterior covariance matrices, thereby reducing the complexity of sampling from the posterior to become effectively *additive* in the combination of tasks of data points, i.e. $\mathcal{O}(n^3 + t^3)$, as compared to $\mathcal{O}(n^3 t^3)$. Our implementation of Matheron's rule draws from the *exact* posterior distribution and does not require random features or inducing points, unlike decoupled sampling [59]. More specifically, our contributions are as follows:

- We propose an *exact* sampling method for multi-task Gaussian processes that has additive time costs in the combination of tasks and data points, rather than multiplicative (Section 3).

- We demonstrate empirically how large-scale sampling from MTGPs can aid in challenging multi-objective, constrained, and contextual Bayesian Optimization problems (Section 4).

- We introduce a method for efficient posterior sampling for the High-Order Gaussian Process (HOGP) model [64], allowing it to be used for Bayesian Optimization (Section 3.2). This advance allows us to more efficiently perform BO on high-dimensional outputs such as images — including optimizing PDEs, optimizing placements of cell towers for cell coverage, and tuning the mirrors of an optical interferometer which optimizes over 65,000 tasks jointly (Section 4.3).

The rest of the paper is organized as follows: First, in Section 2 we review GPs, MTGPs, and sampling procedures from the posterior in both GPs and MTGPs. In Section 3, we review Matheron's rule for sampling from GP posteriors and explain how to employ it for efficient sampling from MTGP models including the HOGP model. In Section 4, we illustrate the utility of our method on a wide suite of problems ranging from constrained BO to the first demonstration of large scale composite BO with the HOGP. Please see Appendix A for discussion of the limitations and broader impacts of our work. Our code is fully integrated into BoTorch, see `https://botorch.org/tutorials/composite_bo_with_hogp` and `https://botorch.org/tutorials/composite_mtbo` for tutorials.

## 2 Background

### 2.1 Bayesian Optimization

In Bayesian Optimization (BO), the goal is to minimize an expensive-to-evaluate black-box function, i.e., finding $\min_{x \in \mathcal{X}} f(x)$, by constructing a *surrogate model* to emulate that function. Gaussian processes (GPs) are often used as surrogates due to their flexibility and well-calibrated uncertainty estimates. BO optimizes an *acquisition function* defined on the predictive distribution of the surrogate model to select the next point(s) to evaluate on the true function. These acquisition functions are often written as intractable integrals that are typically evaluated using Monte Carlo (MC) integration

[60, 3]. MC acquisition functions rely on posterior samples from the surrogate model, which should support fast sampling capabilities for efficient optimization [4]. BO has been applied throughout machine learning, engineering, and the sciences, and many extensions to the setting described above exist. We focus on multi-task BO (MTBO), where $f(x)$ is composed of several correlated tasks [55, 15].

There are many sub-classes of MTBO problems: constrained BO uses surrogate models to optimize an objective subject to black-box constraints [29, 25, 30], contextual BO models a single function that varies across different contexts or environments [38, 11, 26], multi-objective BO aims to explore a Pareto frontier across several objectives [36, 37, 23, 24, 17], and composite BO considers the setting of a differentiable objective function defined on the outputs of a vector-valued black-box function [56, 3, 4]. In all of these problems, the setting is similar: several outputs are modelled by the surrogate, whether the output is a constraint, another objective, or a separate context. As the outputs are often correlated, multi-task Gaussian processes, which model the relationships between the outputs in a data-efficient manner, are a natural and common modeling choice.

## 2.2 Gaussian Processes

**Single Output Gaussian Processes:** We briefly review single output GPs, see Rasmussen and Williams [48] for a more detailed introduction. We assume that $y = f(x) + \varepsilon$, $f \sim \mathcal{GP}(0, k_\theta(x, x'))$, and $\varepsilon \sim \mathcal{N}(0, \sigma^2)$, where $f$ is the noiseless latent function and $y$ are noisy observations of $f$ with standard deviation $\sigma$. $k_\theta(x, x')$ is the kernel with hyperparameters $\theta$ (we will drop the dependence on $\theta$ for simplicity); we use $K_{AB} := k_\theta(A, B)$ to refer to the evaluated kernel function on data points $A$ and $B$, a matrix which has size $|A| \times |B|$. The predictive distributions over new data points, $x_{\text{test}}$, is given by the conditional distribution of the Gaussian distribution. That is, $p(f(x_{\text{test}})|\mathcal{D}, \theta) = \mathcal{N}(\mu^*_{f|\mathcal{D}}, \Sigma^*_{f|\mathcal{D}})$, where $\mathcal{D} := \{X, \mathbf{y}\}$ is the training dataset of size $n = |X|$ and

$$\mu^*_{f|\mathcal{D}} = K_{x_{\text{test}}X}(K_{\text{train}} + \sigma^2 I)^{-1}\mathbf{y}, \tag{1}$$

$$\Sigma^*_{f|\mathcal{D}} = K_{x_{\text{test}}x_{\text{test}}} - K_{x_{\text{test}}X}(K_{\text{train}} + \sigma^2 I)^{-1}K_{Xx_{\text{test}}}, \tag{2}$$

with $K_{\text{train}} := K_{XX}$. For simplicity, we will drop the subscripts $f|\mathcal{D}$ in all future statements. Computing the predictive mean $\mu^*$ and variance $\Sigma^*$ requires $\mathcal{O}(n^3)$ time and $\mathcal{O}(n^2)$ space when using Cholesky decompositions for the linear solves [48]. Sampling is usually performed by

$$f(\mathbf{x}_{\text{test}})|(Y = y) = \mu^* + (\Sigma^*)^{1/2}z, \tag{3}$$

where $z \sim \mathcal{N}(0, I)$. Computing $s$ samples at $n_{\text{test}}$ test points from the predictive distribution costs $\mathcal{O}(n^3 + sn_{\text{test}}^2 + n^2 n_{\text{test}} + n_{\text{test}}^3)$, computed by adding up the cost of all of the matrix vector multiplications (MVMs) and matrix solves. For fixed $\mathbf{x}_{\text{test}}$ we can incur the cubic terms only once by re-using Cholesky factorizations of $K_{\text{train}} + \sigma^2 I$ and $\Sigma^*$ for each sample.

To reduce the time complexity, we can replace all matrix solves with $r < n$ steps of conjugate gradients (CG) and the Cholesky decompositions with rank $r < n$ Lanczos decompositions (an approach called LOVE [45]). These change the major scaling from $n^3$ down to $rn^2$ and the overall time complexity to $\mathcal{O}(rn^2 + srn_{\text{test}} + rnn_{\text{test}} + rn_{\text{test}}^2)$ [45, 28]. In general, $r \ll n$ is used and is usually accurate to nearly numerical precision [28]. We provide additional details in Appendix B.

**Multi-Output Gaussian Processes:** One straightforward way of modelling multiple outputs is to consider each output as an independent GP, modelled in batch together, either with shared or independent hyperparameters [48, 28, 25]. However, there are two major drawbacks to this approach: (i) the model is not able to model correlations between the outputs, and (ii) if there are many outputs then maintaining a separate model for each can result in high memory usage and slow inference times. To remedy these issues, Higdon et al. [33] propose the PCA-GP, using principal component analysis (PCA) to project the outputs to a low-dimensional subspace and then use batch GPs to model the lower-dimensional outputs.

We consider **multi-task Gaussian processes** (MTGPs) with the intrinsic co-regionalization model (ICM), which considers the relationship between responses as the product of the data and task features [32, 6, 1]. We focus on this model due to its popularity and simplicity, leaving a similar derivation of the linear model of co-regionalization to Appendix C.1.1. Given inputs $x$ and $x'$ belonging to tasks $i$ and $j$, respectively, the covariance under the ICM is $k([x, i], [x', j]) = k_D(x, x')k_t(i, j)$. Given $n$ data points $X$ with the $n$ associated task indices $\mathcal{I}$, the covariance is a Hadamard product of the data

Table 1: Time complexities for posterior sampling in single-output, multi-task, and high-order Gaussian Process (HOGP) models. Time complexities shown in blue are our contributions that have not yet been considered by the literature. Standard sampling from MTGPs scales multiplicatively in the combination of the number of tasks, $t$, and the number of data points, $n$, while using Matheron's rule reduces the combination to effectively become additive in these components.

| Model | Distributional (Standard) (Eq. 3) | With Matheron's rule (Eq. 5) |
|---|---|---|
| Single-Output | $\mathcal{O}(n^3 + n_{\text{test}}^3)$ | $\mathcal{O}(n^3 + n_{\text{test}}^3)$ |
| Multi-Task | $\mathcal{O}((n^3 + n_{\text{test}}^3)t^3)$ | $\mathcal{O}((n^3 + n_{\text{test}}^3) + t^3)$ |
| HOGP | $\mathcal{O}((n^3 + n_{\text{test}}^3)\prod_{i=2}^{d} d_i^3)$ | $\mathcal{O}((n^3 + n_{\text{test}}^3) + \sum_{i=2}^{k} d_i^3)$ |

kernel and the task kernel, $K_{\text{train}} = K_{XX} \odot K_{\mathcal{II}}$, and the response $\mathbf{y}$ is still of size $n$. We term this implementation of multi-task GPs "Hadamard MTGPs" in our experiments. In general, there is no easily exploitable structure in this model.

If we observe each task at each data point (the so-called *block design* case), the covariance matrix becomes Kronecker structured, e.g. $K_{\text{train}} = K_{XX} \otimes K_T$, where $K_{XX}$ is the data covariance matrix and $K_T$ is the $t \times t$ task covariance matrix between tasks [6], and we now have $nt$ scalar responses. To simplify our exposition, we assume that $K_T$ is full-rank (this is not required as we can use pseudo-inverses in place of inverses). Thus, the GP prior is $\text{vec}(\mathbf{y}) \sim \mathcal{N}(0, K_{XX} \otimes K_T)$, where $\mathbf{y}$ is a matrix of size $n \times t$, and $\text{vec}(\mathbf{y})$ is a vector of shape $nt$. The GP predictive distribution is given by $p(f^*|x_{\text{test}}, \mathcal{D}) = \mathcal{N}(\mu^*, \Sigma^*)$, where

$$\mu^* = (K_{x_{\text{test}},X} \otimes K_T)(K_{XX} \otimes K_T + \sigma^2 I_{nT})^{-1} \text{vec}(\mathbf{y}),$$

$$\Sigma^* = (K_{x_{\text{test}},x_{\text{test}}} \otimes K_T) - (K_{x_{\text{test}},X} \otimes K_T)(K_{XX} \otimes K_T + \sigma^2 I_{nT})^{-1}(K_{x_{\text{test}},X}^\top \otimes K_T). \quad (4)$$

The kernel matrix on the training data, $K_{XX} \otimes K_T$, is of size $nt \times nt$, which under standard (Cholesky-based) approaches yields inference cubic and multiplicative in $n$ and $t$, that is $\mathcal{O}(n^3 t^3)$. However, the Kronecker structure can be exploited to compute the posterior mean and variance in $\mathcal{O}(nt(n+t) + n^3 + t^3)$, which is dominated by the individual cubic terms [50, 54].

Sampling from the posterior distribution in (4) produces additional computational challenges as we must compute a root (e.g. Cholesky) decomposition of $\Sigma^*$, which naively costs $\mathcal{O}((n_{\text{test}}t)^3)$ plus an additional Cholesky decomposition of $(K_{XX} \otimes K_T + \sigma^2 I)^{-1}$, which similarly costs $\mathcal{O}((nt)^3)$ time [6]. Thus, the time complexity of drawing $s$ samples is *multiplicative* in $n$ and $t$, $\mathcal{O}((nt)^3 + (n_{\text{test}}t)^3 + s((nt)^2 + (n_{\text{test}}t)^2))$. Using CG and LOVE reduces the complexity; see Appendix B.4.

**High-Order Gaussian Processes:** Recently, Zhe et al. [64] proposed the high-order Gaussian process (HOGP) model, a MTGP designed for matrix- and tensor-valued outputs. Given outputs $\mathbf{y} \in \mathbb{R}^{d_1 \times \cdots \times d_k}$ (e.g. a matrix or tensor), the covariance is the product of the data dimension and each output index ($i_l$ and $i_l'$ respectively) $k([x, i_1, \cdots, i_k], [x', i_1', \cdots, i_k']) = k(x, x')k(v_{i_1}, v_{i_1'}') \cdots k(v_{i_k}, v_{i_k'}')$, where $i_1, \cdots, i_k$ are the indices for the output tensor and $v_1, \cdots, v_k$ are latent parameters that are optimized along with the kernel hyper-parameters. Thus, $K_T$ in the MTGP framework is replaced by a chain of Kronecker products, so that the GP prior is $\text{vec}(\mathbf{y}) \sim \mathcal{N}(0, K_{XX} \otimes K_2 \otimes \cdots \otimes K_k)$. Exploiting the Kronecker structure, computation of posterior mean, posterior variance and hyper-parameter learning takes $\mathcal{O}(n^3 + \sum_{i=2}^{k} d_i^3 + n \prod_{i=1}^{k} d_i)$ time as $d_1 = n$. Zhe et al. [64] demonstrate that the HOGP outperforms PCA-GPs [33], among other high-dimensional output GP methods, with respect to prediction accuracy. However, their experiments measure only the error of the predictive mean, rather than quantities that use the predictive variance (such as the negative log likelihood or calibration), and they do not provide a way to sample from the posterior distribution.

## 2.3 Matheron's Rule for Single-Task Gaussian Processes

Matheron's rule provides an alternative way of sampling from GP posterior distributions: rather than decomposing the GP predictive covariance for sampling, one can jointly sample from the prior distribution over both train and test points and then update the training samples using the observed data. Matheron's rule is well known in the geostatistics literature where it is used for "prediction by conditional simulation" [14, 13]. Wilson et al. [59] revitalized interest in Matheron's rule within the

machine learning community by developing a decoupled sampling approach that exploits Matheron's rule to use both random Fourier features (RFFs) and inducing point approaches in the context of sampling from the posterior in single task GPs. Wilson et al. [61] extended decoupled sampling to use combinations of RFFs, inducing points, and iterative methods. They applied these approaches to approximate Thompson sampling, simulations of dynamical systems, and deep GPs.

Matheron's rule states that if two random variables, $X$ and $Y$, are jointly Gaussian, then

$$X|(Y = y) \stackrel{d}{=} X + \text{Cov}(X,Y)\text{Cov}(Y,Y)^{-1}(y - Y),$$

where $\text{Cov}(a, b)$ is the covariance of $a$ and $b$ and $\stackrel{d}{=}$ denotes equality in distribution [20, 59]. The identity can easily be shown by computing the mean and variance of both sides. Following Wilson et al. [59], we can use this identity to draw posterior samples

$$f^*|(Y + \epsilon = y) \stackrel{d}{=} f^* + K_{x_{\text{test}}X}(K_{XX} + \sigma^2 I)^{-1}(y - Y - \epsilon), \tag{5}$$

where $\epsilon \sim \mathcal{N}(0, \sigma^2 I)$, $f^*$ is the random variable and $f^*|(Y + \epsilon = y)$ is the conditional random variable we are interested in drawing samples from. To implement this procedure, we first draw samples from the joint prior (of size $n + n_{\text{test}}$):

$$(f, Y) \sim \mathcal{N}\left(0, \begin{pmatrix} K_{XX} & K_{x_{\text{test}}X} \\ K_{Xx_{\text{test}}} & K_{x_{\text{test}}x_{\text{test}}} \end{pmatrix}\right). \tag{6}$$

For shorthand, we denote the joint covariance matrix in (6) by $\mathcal{K}_{\text{joint}}$. We next sample $\bar{\epsilon} \sim \mathcal{N}(0, \sigma^2 I)$ and compute: $\bar{f} = f + K_{x_{\text{test}}X}(K_{XX} + \sigma^2 I)^{-1}(y - Y - \bar{\epsilon})$. The solve is against a matrix of size $n \times n$ so that $\bar{f}$ is our realization of the random variable $f^*|(Y + \epsilon = y)$. The total time requirements are then $\mathcal{O}((n + n_{\text{test}})^3 + n^3)$, which is slightly slower than $\mathcal{O}(n_{\text{test}}^3 + n^3)$. Thus, sampling from the single-task GP posterior using (5) is slower than the standard method based on (1) and (2).

## 3 Matheron's Rule for Multi-Task Sampling

While the time complexity of Matheron-based posterior sampling is inferior for single-task GPs, the opposite holds true for multi-task GPs, provided that one exploits the special structure in the covariance matrices. In the following, we demonstrate the advantages in using Matheron-based MTGP sampling in Section 3.1, extend it to the HOGP [64] in Section 3.2, and identify further advantages of it for Bayesian Optimization in Section 3.3. This approach allows for further pre-computations than distributional sampling and that it maintains the same convergence results.

### 3.1 Extending Matheron's Rule for Multi-task GPs

The primary bottleneck that we wish to avoid is the multiplicative scaling in $n$ (or $n_{\text{test}}$) and $t$. Ideally, we hope to achieve posterior sampling that is additive in $n$ and $t$, similar to how Kronecker matrix vector multiplication is additive in its components. Unlike in the single task case, Matheron's rule can substantially reduce the time and memory complexity sampling for MTGPs. For brevity we limit our presentation here to the core ideas, please see Appendix C for further discussion of the implementation, as well as Appendix B.1 for a description of the Kronecker identities we exploit.

The covariance $\mathcal{K}_{\text{joint}}$ in (6) is structured as the Kronecker product of the joint test-train covariance matrix and the inter-task covariance; that is, $\mathcal{K}_{\text{joint}} = K_{(X,x_{\text{test}}),(X,x_{\text{test}})} \otimes K_T$, where $K_{(X,x_{\text{test}}),(X,x_{\text{test}})}$ appends $n_{\text{test}}$ rows and columns to the joint training data covariance matrix $K_{XX}$. To sample from the prior distribution, we need to compute a root decomposition of $\mathcal{K}_{\text{joint}}$.

We assume that we have pre-computed $RR^\top = K_{XX}$ with either a Cholesky decompostion ($\mathcal{O}(n^3)$ time) or a Lanczos decomposition ($\mathcal{O}(rn^2)$ time). Then, we update $R$ to compute $\tilde{R}\tilde{R}^\top \approx K_{(x_{\text{test}},X),(x_{\text{test}},X)}$. Chevalier et al. [12] used a similar strategy to update samples in the context of single task kriging. Following Jiang et al. [34, Prop. 2], computing $\tilde{R}$ from $R$ costs $\mathcal{O}(rn_{\text{test}}n + rn_{\text{test}}^2)$ time, dominated by the $rn_{\text{test}}n$ terms for small $n_{\text{test}}$. Using a Cholesky decomposition, this is $\mathcal{O}(n_{\text{test}}n^2)$ time following the same procedure. We then have

$$\mathcal{K}_{\text{joint}} = K_{(x_{\text{test}}X),(x_{\text{test}}X)} \otimes K_T = (\tilde{R} \otimes L)(\tilde{R} \otimes L)^\top, \tag{7}$$

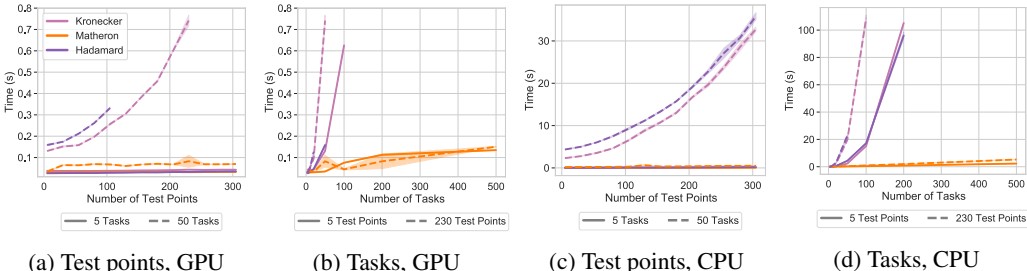

|     |     |     |     |
|:---:|:---:|:---:|:---:|
| (a) Test points, GPU | (b) Tasks, GPU | (c) Test points, CPU | (d) Tasks, CPU |

Figure 2: Timings for distributional sampling with Hadamard and Kronecker MTGPs as well as Matheron's rule sampling for a MTGPs as the number of test points vary for fixed tasks **(a,c)** and as the number of tasks vary for fixed test points **(b,d)** on a single Tesla V100 GPU **(a,b)** and on a single CPU **(c,d)**. The multiplicative scaling of the number of tasks and data points creates significant timing and memory overhead, causing the Kronecker and Hadamard implementations to run out of memory very quickly for all but the smallest numbers of tasks and data points, whereas sampling using Matheron's rule is efficient even in the many-task large-data regime. The plots show mean and two standard errors over 10 trials on the GPU, and 6 trials on the CPU.

where $LL^\top = K_T$. To use the root to sample from the joint random variables, $(f, Y)$, we need to compute only $(f, Y) = (\tilde{R}^\top \otimes L^\top)z$, where $z \sim \mathcal{N}(0, I_{nt})$; this computation is a Kronecker matrix vector multiplication (MVM), which costs just $\mathcal{O}(nt(n + t))$ time. Thus, the overall cost of sampling to compute the joint random variables is $\mathcal{O}(n^3 + t^3 + nt(n + t) + n_{\text{test}}n^2)$ time, reduced to $\mathcal{O}(rn^2 + rt^2 + rnt + r^2t + rn_{\text{test}}n)$ if using Lanczos decompositions throughout. $L$ only needs to be computed once, so samples at new test points only require re-computing $\tilde{R}$ and further MVMs.

Computing the solve $w = (K_{XX} \otimes K_T + \sigma^2 I_{nT})^{-1}(y - Y - \epsilon)$ takes $\mathcal{O}(n^3 + t^3 + nt(n + t))$ time using eigen-decompositions and Kronecker MVMs. The cost of the eigen-decomposition dominates the Kronecker MVM cost so the major cost in this is $\mathcal{O}(n^3 + t^3)$. Finally, there is one more Kronecker MVM $(K_{x_{\text{test}}, X} \otimes K_T)w$ which takes $\mathcal{O}(nt^2 + n_{\text{test}}nt)$ time. We then only need to reshape $\bar{f}$ to be of the correct size.

Therefore, the total time complexity of using Matheron's rule to sample from the MTGP posterior is $\mathcal{O}(n^3 + t^3)$ for small $n_{\text{test}}$, as the cubic terms dominate due to the Cholesky and eigen-decompositions. We show the improvements from using Matheron's rule in Table 1, where the dominating terms are now *additive in the combination of the tasks and the number of data points, rather than multiplicative.* Memory complexities, which are also much reduced, are provided in Table 2 in Appendix C. Finally, we emphasize that sampling in this manner is *exact* up to floating point precision as the solves we use are all exact by virtue of being computed with eigen-decompositions.

### 3.2 Extension to HOGP

The HOGP model can be seen as a special case of the general procedure for sampling MTGPs. We replace $K_T$ with kernels across each tensor dimension of the response, so that $K_T = \otimes_{i=2}^k K_i$. Therefore, the time complexities for sampling go from a cubic dependence, $n^3 \prod_{i=2}^k d_i^3$, to a $(n \prod_{i=1}^k d_i) + (n^3 + \sum_{i=2}^k d_i^3)$ dependence. The latter will usually be dominated by the additive terms for $k \lesssim 5$, as generally $n$ is much larger than the tensor sizes, hence their product will generally be less than $n^2$. Prior to this work, sampling from the posterior was infeasible for all but the smallest HOGP models due to the time complexity. By using Matheron's rule, we can sample from HOGP posterior distributions over large tensors, unlocking the model for a much wider range of applications such as BO with composite objectives computed on high-dimensional simulator outputs.

### 3.3 Usage in Bayesian Optimization

The primary usage of efficient multi-task posterior sampling that we consider is that of Bayesian optimization. Here, we want to use and optimize Monte Carlo acquisition functions that require many samples from the posterior of the surrogate model.

At a high level, Bayesian optimization loops look like the following procedure that we repeat after initializing several random points. First, we fit MTGPs by maximizing the log marginal likelihood (see Appendix B.3) Then, we draw many posterior samples from the MTGP in the context of computing MC acquisition functions, e.g. (A.5). We use gradient-based optimization to find the points $x$ which optimize the acquisition $\hat{\alpha}_N(x; y)$. After choosing these points, $x_{\text{cand}}$, we then query the function, returning $y_{\text{cand}}$ and updating our data with these points. Finally, we continue back to the top of the loop, and re-train the MTGP.

**Convergence Results:** The convergence guarantees for Sample Average Approximation (SAA) of MC acquisition functions from Balandat et al. [4] still apply if posterior samples are drawn via Matheron's rule. Consider acquisition functions of the form $\alpha(x; y) = \mathbb{E}\big[h(f(x)) \mid Y = y\big]$ with $h : \mathbb{R}^{n_{\text{test}} \times t} \to \mathbb{R}$, and their MC approximation $\hat{\alpha}_N(x; y) := \frac{1}{N} \sum_{i=1}^{N} a(g(f_i^*))$, where $f_i^*$ are i.i.d. samples drawn from the model posterior at $x \in \mathbb{R}^{n_{\text{test}}}$ using Matheron's rule. Then, under sufficient regularity conditions, the optimizer $\arg\max_x \hat{\alpha}_N(x; y)$ converges to the true optimizer $\arg\max_x \alpha(x; y)$ almost surely as $N \to \infty$. For a more formal statement and proof of this result see Appendix C.3.

## 4 Experiments

We first demonstrate the computational efficiencies gained by posterior sampling using Matheron's rule. While this contribution is much more broadly useful, here we focus on the benefits it provides for BO. Namely, we show that accounting for correlations between outcomes in the model improves performance in multi-objective and large-scale constrained optimization problems. Finally, we perform composite BO with tens of thousands of tasks using HOGPs with Matheron-based sampling. Additional experiments on contextual policy optimization are given in Appendix D.2. All plotted lines represent the mean over repeated trials with shading represent two standard deviations of the mean across trials.

### 4.1 Drawing Samples from Multi-Task Models

To demonstrate the performance gains of sampling using Matheron's rule, we first vary the number of test points and tasks for a fixed number of samples and training points. Following Feng et al. [26], we consider a multi-task version of the Hartmann-6 function, generated by splitting the response surface into tasks using slices of the last dimension. In Figures 2a and 2c, we vary the number of test points for 5 and 50 tasks, demonstrating that Matheron's rule sampling is faster and more memory efficient than distributional sampling with either Kronecker or Hadamard MTGPs. In Figures 2b and 2d, we vary the number of tasks for fixed test points, again finding that distributional sampling is only competitive for 5 tasks on the GPU. See Appendix D for sampling with LOVE predictive covariances.

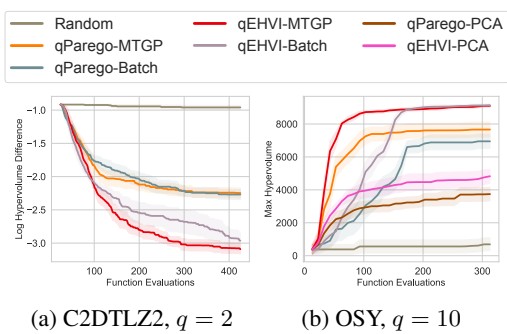

(a) C2DTLZ2, $q = 2$     (b) OSY, $q = 10$

Figure 3: Constrained multi-objective Bayesian Optimization tasks. MTGPs outperform batch models in both the **(a)** small batch ($q = 2$) and the **(b)** large batch ($q = 10$) setting. The latter was previously computationally infeasible for MTGPs.

### 4.2 Multi-Objective Bayesian Optimization

We next consider constrained multi-objective BO, where the goal is to find the *Pareto Frontier*, i.e., the set of objective values for which no objective can be improved without deteriorating another while satisfying all constraints. To measure the quality of the Pareto Frontier, we compute the hypervolume (HV) of the non-dominated objectives [62]. The optimization problem is made more difficult by the presence of black-box constraints (and hence additional outcomes) that must be modelled. We use MC batch versions of the ParEGO and EHVI acquisition functions (qParEGO and

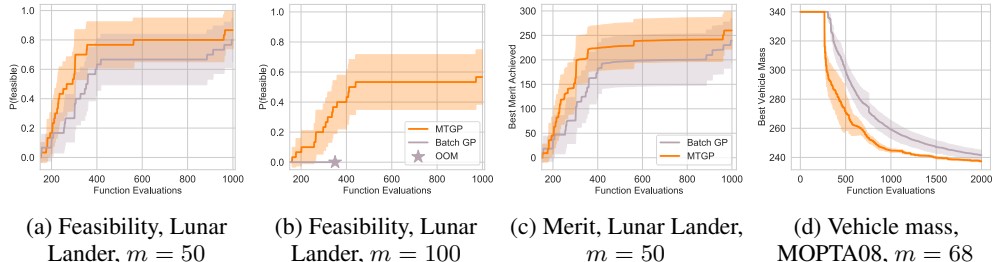



(a) Feasibility, Lunar Lander, $m = 50$     (b) Feasibility, Lunar Lander, $m = 100$     (c) Merit, Lunar Lander, $m = 50$     (d) Vehicle mass, MOPTA08, $m = 68$



Figure 4: Scalable constrained Bayesian Optimization on Lunar Lander $m = 50, 100$ **(a-c)** and on MOPTA08 **(d)**. On all three problems, a multi-task GP provides better solutions with better data efficiency. The batch GP reaches feasibility on lunar lander with 50 constraints **(a)** and a competitive solution **(c)** but requires more trials, while on 100 constraints, it simply runs out of memory while the MTGP succeeds. On MOPTA08, the MTGP reaches a better solution faster **(d)**.

qEHVI) [17]. To generate new candidate points, qParEGO maximizes expected improvement using a random Chebyshev scalarization of the objectives, while qEHVI maximizes expected hypervolume improvement. As far as we are aware, Shah and Ghahramani [52] are the only authors to investigate the use of MTGPs in combination with multi-objective optimization, but only consider 2-4 tasks in the sequential setting (generating one point at a time). Here, we use full rank inter-task covariances with LKJ priors [40] which we find to work well even in the low data regime. Our Matheron-based sampling scales to large batches and tasks, and is more sample-efficient on both tasks.

We compare Matheron sampled MTGPs to batch independent MTGPs on the C2DTLZ2 [19] (2 objectives, 1 constraint for a total of 3 modelled tasks) and OSY [43] (2 objectives, 6 constraints for a total of 8 modelled tasks) test problems. On OSY, we also compare to PCA GPs [33] due to the larger number of outputs. Following Daulton et al. [17] we use both qParEGO and qEHVI with $q = 2$, for C2DTLZ2 and optimize for 200 iterations. In Figure 3a, we see that the MTGPs outperform their batch counterparts by coming closer to the known optimal HV. On OSY, in Figure 3b, we plot the maximum HV achieved for each method, using a batch size of $q = 10$, optimizing for 30 iterations, where again we see that the MTGPs significantly outperform their batch counterparts as well as the PCA GPs, which stagnate quickly.

## 4.3 Scalable Constrained Bayesian Optimization

We next extend scalable constrained Bayesian Optimization [SCBO, 25], a state of the art algorithm for constrained BO in high-dimensional problems, to use MTGPs instead of independent GPs. In constrained BO, the goal is to minimize the objective, $f$, subject to black box constraints, $c_i$; e.g.,

$$\arg\min_x f(x) \quad \text{s.t.} \quad c_i(x) \leq 0, \quad \forall i \in \{1, \cdots, m\}. \tag{8}$$

SCBO is a method based on trust regions and uses batched independent GPs to model the outcome and transformed constraints. We compare to their results on their two largest problems — the 12-dimensional lunar lander and the 124-dimensional MOPTA08 problem, using the same benchmark hyper-parameters. To extend their approach, we replace the independent GPs with a single MTGP model with a full rank ICM kernel over objectives and constraints.

**Lunar Lander:** We first consider the lunar lander problem with both 50 and 100 constraints from the OpenAI Gym [8]. Following Eriksson and Poloczek [25], we initialize with 150 data points and use TuRBO with Thompson sampling with batches of $q = 20$ for a total of 1000 function evaluations and repeat over 30 trials. Using a multi-task GP reduces the number of iterations to achieve at least a $50\%$ chance of feasibility by about 75 steps for the 50 constraint problem (Figure 4a). On the 100 constraint problem, the batch GP runs out of memory after 350 steps on a single GPU and never achieves feasibility, as indicated in Figure 4b. In Figure 4c, we show the best merit $(f(x) \prod_{i=1}^{m} 1_{c_i(x) \leq 0})$ achieved where the MTGPs are able to achieve feasibility in fewer samples, but do not reach significantly higher reward. Wall clock times for the $m = 50$ constraint problem, a table of the steps to achieve feasibility, and a comparison to PCA-GPs [33] are given in Appendix D.

**MOPTA08:** We next compare to batch GPs on the MOPTA08 benchmark problem [35] which has 68 constraints that measure the feasibility of a vehicle's design, while each dimension involves gauges,

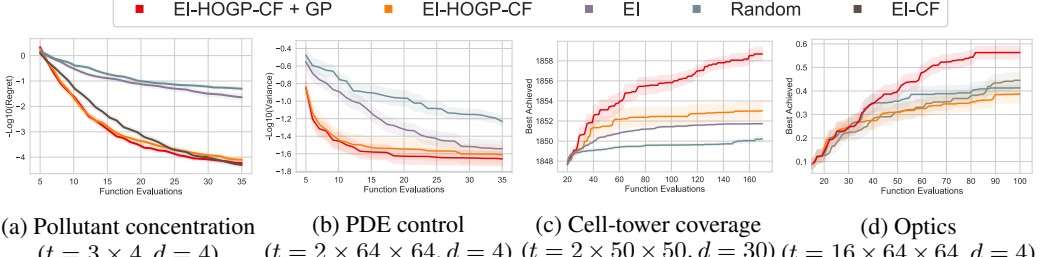

(a) Pollutant concentration  
$(t = 3 \times 4, d = 4)$

(b) PDE control  
$(t = 2 \times 64 \times 64, d = 4)$

(c) Cell-tower coverage  
$(t = 2 \times 50 \times 50, d = 30)$

(d) Optics  
$(t = 16 \times 64 \times 64, d = 4)$

Figure 5: Performance of BO with and without HOGP-based composite objectives (EI-HOGP-CF) on four scientific problems. EI-HOGP-CF outperforms a standard BO model directly on the metric itself (EI) and a random baseline (Random). Composite BO with the HOGP also outperforms composite BO with independent GPs (EI-CF). Our smooth latents for the HOGP (EI-HOGP + GP) typically outperform the HOGP itself. EI-CF is only feasible on the smallest problem.

materials, and shapes. We use Thompson sampling to acquire points with a batch size of $q = 10$, 130 initial points, and optimize for 2000 iterations repeating over 9 trials. The results are shown in Figure 4d, where we again observe that SCBO with MTGPs significantly improves both the time to feasibility and the best overall objective found. Using MTGPs would have been computationally infeasible in this setting without our efficient posterior sampling approach.

### 4.4 Composite Bayesian Optimization with the HOGP

Finally, we push well beyond current scalability limits by extending BO to deal with *many thousands* of tasks, enabling us to perform sample-efficient optimization in the space of images. *Composite BO* is a form of BO where the objective is of the form $\max_x g(h(x))$, where $h$ is a multi-output black-box function modelled by a surrogate and $g$ is cheap to evaluate and differentiable. Decomposing a single objective into constituent objectives in this way can provide substantial improvements in sample complexity [3]. Balandat et al. [4] gave convergence guarantees for optimizing general sampled composite acquisition function objectives that extend to MTGP models under the same regularity conditions. However, both works experimentally evaluate only batch independent multi-output GPs.

We compare to three different baselines: random points (Random), expected improvement on the metric (EI), and batch GPs optimizing EI in the composite setting (EI-CF). We consider the HOGP [64] with Matheron's rule sampling (HOGP-CF) as well as an extension of HOGPs with a prior over the latent parameters that encourages smoothly varying latent dimensions (HOGP-CF + GP); see Appendix C.2 for details. More detailed descriptions of the problems are provided in Appendix D.

**Chemical Pollutants:** Following Astudillo and Frazier [3], we start with a simple spatial problem in which environmental pollutant concentrations are observed on a $3 \times 4$ grid originally defined in Bliznyuk et al. [5]. The goal is to optimize a set of four parameters to achieve the true observed value by minimizing the mean squared error of the output grid to the output grid of the true parameters. The results, over 50 trials, are shown in Figure 5a, where we find that the HOGP models with these few tasks outperform both independent batch GPs (but slightly) and BO on the metric itself.

**Optimizing PDEs:** As a larger experimental problem, we consider optimizing the two diffusivity and two rate parameters of a spatial Brusselator PDE (solved in py-pde [65]) to minimize the weighted variance of the PDE output as an example of control of a dynamical system. Here, we solve the PDE on $64 \times 64$ grid, producing output solutions of size $2 \times 64 \times 64$. Results over 20 trials are shown in Figure 5b, where the HOGP models outperform EI fit on the metric and the random baseline.

**Cell-Tower Coverage:** Following Dreifuerst et al. [21], we optimize the simulated "coverage map" resulting from the transmission power and down-tilt settings of $15$ cell towers (for a total of $30$ parameters) based on a scalarized quality metric combining signal power and inference at each location so as to maximize total coverage, while minimizing total interference. To reduce model complexity, we down-sample the simulator output to $50 \times 50$, initializing the optimization with $20$ points. Figure 5c presents the results over $20$ trials, where the HOGP models with composite objective EI outperform EI, indicating that modeling the full high-dimensional output is valuable.

**Optical Interferometer:** Finally, we consider the tuning of an optical interferometer by the alignment of two mirrors as in Sorokin et al. [53]. Here, the problem is to optimize the mirror coordinates

to align the interferometer so that it reflects light without interference. There is a sequence of 16 different interference patterns and the simulation outputs are $64 \times 64$ images (a tensor of shape $16 \times 64 \times 64$). Thus, we jointly model 65,536 output dimensions. Scaling composite BO to a problem of this size would be impossible without the step change in scalability our method provides. Results are shown in Figure 5d over 20 trials, where we find that the HOGP-CF + GP models outperform EI, with the HOGP + GP under-performing (perhaps due to high frequency variation in the images).

Across all of our experiments, we consistently find that composite BO is considerably more sample efficient than BO on the metric itself, with significant computational improvements gained from using the HOGP as compared to batch GPs, which are infeasible much beyond batch sizes of 100. Furthermore, our structured prior approach for the latent parameters of the HOGP tends to outperform the random initialization strategy in the original work of Zhe et al. [64].

## 5 Conclusion

We demonstrated the utility of Matheron's rule for sampling the posterior in multi-task Gaussian processes. Combining Matheron's rule with scalable Kronecker algebra enables posterior sampling in $\mathcal{O}(n^3 + t^3)$ rather than the previous $\mathcal{O}(n^3 t^3)$ time. This renders posterior sampling from high-order Gaussian processes [64] practical, for the first time unlocking Bayesian Optimization with composite objectives defined on high-dimensional outputs. This increase in computational efficiency dramatically reduces the time required to do multi-task Bayesian Optimization, and thus enables practitioners to achieve better automated Bayesian decision making. While we focus on the application to Bayesian Optimization in this work, our contribution is much broader and provides a step change in scalability to all methods that in involve sampling from MTGP posteriors. We hope in the future to explore stronger inter-task covariance priors to make MTGP model fits even more sample efficient.

## Acknowledgements

WJM, AGW are supported by an Amazon Research Award, NSF I-DISRE 193471, NIH R01 DA048764-01A1, NSF IIS-1910266, and NSF 1922658 NRT-HDR:FUTURE Foundations, Translation, and Responsibility for Data Science. WJM was additionally supported by an NSF Graduate Research Fellowship under Grant No. DGE-1839302, and performed part of this work during an internship at Facebook. We would like to thank Paul Varkey for providing code and David Eriksson, Qing Feng, and Polina Kirichenko for helpful discussions.

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
