# Supplementary Materials for Bayesian Optimization with High-Dimensional Outputs

## Organization

The Appendix is organized as follows:

- Appendix A describes limitations and negative societal impacts of our work.

- Appendix B describes further background and related work on Kronecker matrix vector products, Matheron's rule, multi-task Gaussian process models, and sampling multi-task posteriors using LOVE [45].

- Appendix C gives a more detailed description of sampling multi-task Gaussian process posteriors using Matheron's rule, our set of priors for the HOGP [64], and a proof of convergence using Matheron's rule sampling to optimize MC acquisition functions.

- Appendix D describes two more experiments on multi-task BO and contextual BO, before giving more detail and results on the experiments in the main paper.

## A    Limitations and Societal Impacts

From a practical perspective, we see several inter-related limitations:

- If the underlying multi-output function we are trying to model has very different lengthscales for each output, then the shared data covariance matrix of the MTGP may not be able to model each output very well. In practice, this seems to be rather rare, but may prove to be more problematic for the HOGP model due to the number of outputs that we model.

- Numerical instability can be an issue when solving systems of equations using eigen-decompositions; however, we did not find it to be problematic as our implementation performs the eigen-decomposition in double precision despite all other computations being performed in single precision.

- As currently described, we can only apply our method to block design, fully observed settings where all data points and tasks are observed at the same time. In future work, we hope to extend past this limitation, perhaps using the approaches of [63, 58].

- *Autokrigeability* may play a larger role here than it does in standard MTGP scenarios, especially when the black box function is observed without observation noise. See Bonilla et al. [6] for a longer description of this problem. However, we still get substantial computational enhancements from using the HOGP and Kronecker structure compared to modeling each output with an independent GP model.

- Negative transfer can arise if the relationship between tasks is highly non-linear. Non-linear MTGP models are needed to remedy this issue, rather than the ICM model we consider [7].

Looking farther out, we do not broadly anticipate direct negative societal impacts as a result of our work. However, Bayesian Optimization, which we focus on in this paper, is a very generic methodology for optimizing black box functions. This technology can be used for good reasons such as public health surveillance and modelling [41, 2] or technological design, such as the radio frequency tower location and optics problems discussed in this paper. These types of applications should hopefully increase the likelihood of deployment of new advanced technologies such as $5G$ cell coverage globally and thus help to provide more people with stable jobs and employment.

# B  Further Background and Related Work

In this Appendix, we note of several more references in the geostatistics community who use Matheron's rule in multi-task Gaussian processes (MTGPs), give more background on MTGPs, before moving into a more detailed description of LOVE predictive variances and fast sampling [45], including for multi-task GPs.

## B.1  An Extended Reference on Kronecker Structure

We may exploit Kronecker structure in matrices in order to perform more efficient matrix vector multiplies and solves. For a more detailed introduction to Kronecker matrices and their properties, please see Saatchi [51, Chapter 5] as well as Sections 1.3.7-8 and 12.3 of Golub and Van Loan [31]. Matrix vector multiplies (MVMs) are efficient and can be computed from:

$$z = (K_1 \otimes K_2)\text{vec}(A) = \text{vec}(K_2 A K_1^\top);$$

if $K_1 \in \mathbb{R}^{n_1 \times n_1}$ and $K_2 \in \mathbb{R}^{n_2 \times n_2}$. As a result, computing $z$ costs $\mathcal{O}(n_1^2 + n_2^2 + n_1 n_2(n_1 + n_2))$ time [31, 12.3].

There are several other useful Kronecker product properties (again summarized from Saatchi [51], Golub and Van Loan [31] amongst other sources): $(A \otimes B)(C \otimes D) = AC \otimes BD$, if the shapes match, and $(K_1 \otimes K_2)^{-1} = (K_1^{-1} \otimes K_2^{-1})$, and $\log |K_1 \otimes K_2| = n_2 \log |K_1| + n_1 \log |K_2|$. Root (cholesky) decompositions also factorize across the Kronecker product as $A \otimes B = LL^\top \otimes RR^\top = (L \otimes R)(L^\top \otimes R^\top)$.

Using these properties, we then are able to compute matrix inverses of Kronecker plus constant diagonal matrices as $(K_1 \otimes K_2 + \sigma^2 I)^{-1} = (Q_1 \otimes Q_2)(\Lambda_1 \otimes \Lambda_2 + \sigma^2 I)^{-1}(Q_1 \otimes Q_2)^\top$, where $K_i = Q_i \Lambda_i Q_i^\top$ (the eigen-decomposition of $K_i$). As eigen-decompositions cost cubic time in the size of the matrix then the total cost for these matrix solves is $\mathcal{O}(n^3 + t^3 + nt(n + t))$ with the final term coming from matrix vector products. In general, there is no efficient way to compute matrix inverses of the form: $(K_1 \otimes K_2 + T_l)^{-1}$ where $T_l$ is a diagonal matrix that is non-constant. One can use conjugate gradients to compute solves in that setting.

However, as Rakitsch et al. [47] demonstrate, if we assume a structured noise term, e.g. a likelihood that is $\vec{Y} \sim \mathcal{N}(f, \Sigma_X \otimes \Sigma_T)$, then there is an efficient method of computing matrix inverses and solves:

$$(K_1 \otimes K_2 + \Sigma_X \otimes \Sigma_T)^{-1} = (Q_X \Lambda_X^{1/2} \otimes Q_T \Lambda_T^{1/2})(\tilde{Q}_1 \otimes \tilde{Q}_2)$$
$$(\tilde{\Lambda}_1 \otimes \tilde{\Lambda}_2 + I)^{-1}(\tilde{Q}_1 \otimes \tilde{Q}_2)^\top(Q_X \Lambda_X^{1/2} \otimes Q_T \Lambda_T^{1/2})^\top,$$

where $\tilde{Q}_1 \tilde{\Lambda}_1 \tilde{Q}_1^\top = \Lambda_X^{-1/2} Q_X^\top K_1 Q_X \Lambda_X^{-1/2}$ and $\tilde{Q}_2$ is defined in the same manner. This costs two eigen-decompositions and several matrix vector multiplications for a total of $\mathcal{O}(n^3 + t^3 + nt(n + t))$ time.

Rakitsch et al. [47] and Bonilla et al. [6] brush the $nt$ terms in scaling under the rug as they are dominated by the cubic time complexity of the eigen-decompositions. We follow this notation in our results.

## B.2  Matheron's Rule

Matheron's rule is well known in the geostatistics literature where it is called "prediction by conditional simulation" [14, 13]. There, it is also known that it can be applied to multi-task Gaussian processes, as described in de Fouquet [18] and mentioned in Emery [22]. Larocque et al. [39] use Matheron's rule to sample in the multi-task setting (termed co-kriging in that literature) and study the uncertainty of the ICM kernel on groundwater use cases. However, they focus only on two to three tasks and do not exploit the Kronecker structure in the multi-task covariances. Doucet [20] gave a didactic explanation of Matheron's rule with the goal of introducing it to the broader machine learning community, explaining its applications in sampling Kalman filters.

## B.3 Multi-task Gaussian Process models

Both computationally efficient, e.g. Bruinsma et al. [9], and variational methods, e.g. Nguyen et al. [42], Dai et al. [16], can be made more efficient in our Matheron's rule implementation. Furthermore, other kernels for MTGPs such as the linear model of coregionalization (LMC) and the semiparametric latent factor models [48] can also be extended to use Matheron's rule in the way that we mention here. See Álvarez et al. [1], Bruinsma et al. [9] for an extended description of the relationships between various models of multi-task GPs. In most of the paper, we focus solely on the exact setting with ICM kernels for both didactic and implementation purposes. We detail the extension to the LMC case in Appendix C.1.1.

**Training Multi-task GPs:**   To train single task GPs, we optimize the marginal log likelihood with gradient based optimization; this approach extends to training multi-task GPs as well. In the single task setting, the marginal log likelihood (MLL) is: $\log p(y) = \frac{1}{2}\left(N\log 2\pi - \log|K + \sigma^2 I| - y^\top(K + \sigma^2 I)^{-1}y\right)$ (Eq. 5.8 in Rasmussen and Williams [48]). To extend the MLL into the multi-task setting, we only need to exploit Kronecker identities as described in Section B.1 and focus solely on the constant diagonal case.[1] The MLL becomes

$$\log p(y) = \frac{1}{2}\left(NT\log 2\pi - \log|K_X \otimes K_T + \sigma^2 I| - \text{vec}(y)^\top(K_X \otimes K_T + \sigma^2 I)^{-1}\text{vec}(y)\right)$$

The log determinant term simplifies to

$$\log|K_X \otimes K_T + \sigma^2 I| = \log|\Lambda_X \otimes \Lambda_T + \sigma^2 I|$$

which is just the determinant of a diagonal matrix. The quadratic form similarly simplifies

$$\text{vec}(y)^\top(K_X \otimes K_T + \sigma^2 I)^{-1}\text{vec}(y) = \text{vec}(y)^\top(Q_X \otimes Q_T)(\Lambda_X \otimes \Lambda_T + \sigma^2 I)^{-1}(Q_X \otimes Q_T)^\top\text{vec}(y)^\top.$$

We then estimate the kernel hyper-parameters with gradient based optimization of the log marginal likelihood, following Stegle et al. [54], Rakitsch et al. [47]. The predictive means and variances of the MTGP are given by Bonilla et al. [6].

## B.4 LOVE Variance Estimates and Sampling Multi-Task Posteriors

To compute $\mu^*$ in (1), we need to solve the linear system, $(K_{XX} + \sigma^2 I)^{-1}\mathbf{y}$; this solution costs $\mathcal{O}(n^3)$ when using the Cholesky decomposition [48]. Recently, Gardner et al. [28] have proposed using preconditioned conjugate gradients (CG) to solve these linear systems in $\mathcal{O}(rn^2)$ time, where $r$ is the number of conjugate gradients steps. Similarly, to compute the predictive variance in (2), we need to solve $n_{\text{test}}$ systems of equations of the form $(K_{XX} + \sigma^2 I)^{-1}K_{X\mathbf{x}_{\text{test}}}$, which would naively cost $\mathcal{O}(n^3 n_{\text{test}})$ time, reduced to $\mathcal{O}(n_{\text{test}}n + n^2 n_{\text{test}})$ time if we have precomputed the Cholesky decomposition of $(K_{XX} + \sigma^2 I)$.

Pleiss et al. [45] propose to additionally use a cached Lanczos decomposition (called LOVE) such that $RR^\top \approx (K_{XX} + \sigma^2 I)^{-1}$ and then simply perform matrix multiplications against $K_{X\mathbf{x}_{\text{test}}}$ to compute the predictive variances. The time complexity of the Lanczos decomposition is also $\mathcal{O}(n^2 r)$ for computing a rank $r$ decomposition. Sampling proceeds similarly by computing a rank $r$ decomposition to $\Sigma$ in (3). The overall time complexity for computing $s$ samples at from the predictive distribution in the exact formulation is reduced to $\mathcal{O}(rn^2 + srn_{\text{test}} + rn_{\text{test}}^2 + n^2 n_{\text{test}})$. These advances in GP inference have enabled exact single-output GP regression on datasets of up to one million data points [57]. Furthermore, Gardner et al. [28], Pleiss et al. [45] demonstrate that $r < n$ while maintaining accuracy up to numerical precision in floating point.

**LOVE for multi-task predictions.**   It is possible to exploit the Kronecker structure in the posterior distribution to enable more efficient sampling than the naive $\mathcal{O}((n_{\text{test}}t)^3)$ approach [28]. $(K_{XX} \otimes K_T + \sigma^2 I_{nT})$ admits an efficient matrix vector multiply (MVM) due to its Kronecker structure — this MVM takes $\mathcal{O}(nt(n + t))$ time, see Appendix B.1. If we use LOVE to additionally decompose the matrix such that $LL^\top \approx (K_{XX} \otimes K_T + \sigma^2 I_{nT})^{-1}$, then computing $L$ that has rank $r$ takes $\mathcal{O}(r(nt(n + t)))$ time but has a storage cost of $nt \times r$, which is multiplicative in the combination of $n$ and $t$. Then, computing $(K_{x_{\text{test}},X} \otimes K_T)L$ takes $\mathcal{O}((n_{\text{test}}nt + nt^2)r)$ time. This represents

---

[1]Please see Rakitsch et al. [47] for the structured noise case.

an improvement over the naive method, but still ends up requiring computing $\Sigma^* = AA^\top = (K_{x_{\text{test}},x_{\text{test}}} \otimes K_T) - (K_{x_{\text{test}},X} \otimes K_T)LL^\top(K_{x_{\text{test}},X}^\top \otimes K_T)$, which is a $n_{\text{test}}t \times n_{\text{test}}t$ matrix, and therefore costs at least $(n_{\text{test}}t)^2$ time to decompose and thus sample. The overall time complexity is then $\mathcal{O}(r(n_{\text{test}}t)^2 + rn(n_{\text{test}} + t^2))$. Finally, the matrix $A$ must be re-computed from scratch for each new test point $x_{\text{test}}$, and only the matrix $L$ can be reused for different test points (as in a Bayesian Optimization loop).

## C  Further Methods

In this Appendix, we start by giving a more detailed derivation of our efficient Matheron's rule implementation for sampling from multi-task Gaussian processes, then we describe a new set of priors for the HOGP model, before closing with a convergence proof of using Matheron's rule in optimizing Monte Carlo acquisition functions.

### C.1  Details of Using Matheron's Rule

We derive only the zero mean case here for simplicity.

Succinctly, to generate $f(x_{\text{test}})|Y = y$ under the ICM, we may draw a joint sample from the prior

$$(f, Y) \sim \mathcal{N}\left(0, \left(\begin{array}{cc} K_{XX} & K_{Xx_{\text{test}}} \\ K_{x_{\text{test}}X} & K_{x_{\text{test}}x_{\text{test}}} \end{array}\right) \otimes K_T\right) \tag{A.1}$$

and then update the sample via an update from computing $(K_{\text{train}} + T_l)^{-1}(y - Y - \epsilon)$. Here, $T_l$ represents the noise likelihood used — it could be either constant diagonal: $\sigma^2 I$, non-constant diagonal with a variance term for each task: $D$, or Kronecker structured itself: $D_n \otimes T_t$, where $T_t$ is a dense matrix. The formula is given as

$$\bar{f} = f + K_{x_{\text{test}}X}(K_{XX} + \sigma^2 I)^{-1}(y - Y - \epsilon). \tag{A.2}$$

The joint covariance matrix, $\mathcal{K}_{\text{joint}}$, in (A.1) is highly structured

$$\mathcal{K}_{\text{joint}} = \left(\begin{array}{cc} K_{XX} & K_{x_{\text{test}}X} \\ K_{Xx_{\text{test}}} & K_{x_{\text{test}}x_{\text{test}}} \end{array}\right) \otimes K_T = \tilde{R}\tilde{R}^\top \otimes LL^\top = (\tilde{R} \otimes L)(\tilde{R} \otimes L)^\top,$$

where $\tilde{R}\tilde{R}^\top \approx K_{(X,x_{\text{test}}),(X,x_{\text{test}})}$ and $LL^\top = K_T$ and we exploit Kronecker structure. To compute $\tilde{R}$, we follow Jiang et al. [34]'s method for fantasization (given in Proposition 2 therein):

$$\left(\begin{array}{cc} K_{XX} & K_{x_{\text{test}}X} \\ K_{Xx_{\text{test}}} & K_{x_{\text{test}}x_{\text{test}}} \end{array}\right) = \left(\begin{array}{cc} R & 0 \\ L_{12} & L_{22} \end{array}\right)\left(\begin{array}{cc} R & 0 \\ L_{12} & L_{22} \end{array}\right)^\top,$$

where $RR^T = K_{XX}$ and $L_{12}^\top = R^{-1}K_{Xx_{\text{test}}}$. To compute $L_{22}$, we have to compute

$$L_{22} = (K_{x_{\text{test}}x_{\text{test}}} - L_{12}L_{12}^\top)^{1/2}.$$

If we assume a rank $r$ decomposition of $K_{XX}$, computed in $\mathcal{O}(n^2r)$ time (e.g. a LOVE decomposition Pleiss et al. [45]), then computing $L_{12}$ costs $\mathcal{O}(n_{\text{test}}rn)$ if we have stored $R^{-1}$ (or $R^+$). Similarly, computing $L_{22}$ costs $\mathcal{O}(n_{\text{test}}^2r)$ time if we use a Lanczos decomposition (for large $n_{\text{test}}$). We could also use contour integral quadrature [46] to compute $L_{22}v$ at the expense of having to re-compute it every time we want to draw a new sample. Sampling then proceeds by computing

$$(f, Y) = \left(\left(\begin{array}{cc} R & 0 \\ L_{12} & L_{22} \end{array}\right) \otimes L\right)z, \tag{A.3}$$

where $z \sim \mathcal{N}(0, I)$. We can then compute $\epsilon \sim \mathcal{N}(0, T_l)$, where $T_l$ is the noise distribution. Typically, $T_l$ will be diagonal so this sampling just requires taking the square root of $T_l$; it could alternatively use a Kronecker structured root decomposition if not diagonal.

We then need to compute

$$w = (K_{XX} + T_l)^{-1}(y - Y - \epsilon),$$

via efficient Kronecker solves as described in Section B.1 — for example, if $T_l$ is a constant diagonal, use the Kronecker eigen-decomposition and add the constant to the eigenvalues. The diagonal solves generally cost $\mathcal{O}(n^3 + t^3 + nt(n + t))$ time, while even $T_l = \Sigma_{TT} \otimes \Sigma_{NN}$, full rank task and data noises, still costs $\mathcal{O}((n^3 + t^3) + nt(n + t))$ as we only need to perform extra matrix multiplications [47]. Finally, we only need to compute a Kronecker matrix vector multiplication, computing $z = K_{x_{\text{test}}X}w$ and $\bar{f} = f + z$. This exploits Kronecker identities and costs $\mathcal{O}(nt(n + t))$.

We choose to mention the $nt$ terms for precision, despite it typically being dropped in the literature due to the solve costs being dominated by the eigen-decompositions of training and task covariance matrices [47, 33, 6]. In all cases, the $nt$ terms come solely from the Kronecker matrix vector multiplications. The overall time complexity of the operations is then $\mathcal{O}(n^3 + t^3 + nt(n + t))$ time, which is $\mathcal{O}(n^3 + t^3)$ time.

The non-zero mean case can be implemented by adding in the mean function into the joint sample at (A.1) and again at the end of (A.2).

The extension to the HOGP model proceeds like the general ICM case if we replace $L$ by the root decomposition of the kernels across all tensors, $L = \otimes_{i=2}^d L_i$ such that $\otimes_{i=2}^d K_i = \otimes_{i=2}^d L_i L_i^\top$. Again, we only need to update the root decomposition on the data covariance and can re-use the root decomposition on the latent covariances.

Overall memory complexities are shown in Table 2; we ignore the fixed constant train decomposition costs of $K_{XX}$ and/or $K_{XX} + \sigma^2 I$. For single output GPs, this is a constant $\mathcal{O}(n^2)$ ($\mathcal{O}(nr)$ if LOVE is used). For multi-task GPs, it becomes $\mathcal{O}(n^2 + t^2 + nt)$ (or $\mathcal{O}(ntr)$). For the HOGP, it is $\mathcal{O}(\sum_{i=1}^k d_i^2 + \prod_{i=1}^k d_i)$ (or $\mathcal{O}(r(\sum_{i=1}^k d_i))$). The multiplicative scaling in memory is the cost of a single vector (the eigenvalues of $K_{XX}$) for the HOGP and the MTGPs. Unfortunately, combining Lanczos partial eigen-decompositions does not help reduce the memory by as much in the MTGP or HOGP setting due to the necessity of some zero-padding.

Table 2: Memory complexities after pre-computation for posterior sampling in single-output, multi-task, and high-order (HOGP) Gaussian Process models. Matheron's rule allows decomposition across the Kronecker product of the train and task covariances, enabling significant improvements in memory scaling. We ignore pre-computation costs, while the multiplicative terms are single vectors.

| Model | Distributional (Standard) (3) | With Matheron's rule (5) |
|---|---|---|
| Single-Output | $n_{\text{test}}^2$ | $n_{\text{test}}^2 + nn_{\text{test}}$ |
| Multi-Task | $(n_{\text{test}}t)^2$ | $n_{\text{test}}^2 + t^2 + nt$ |
| HOGP | $(n_{\text{test}}^2)\prod_{i=2}^d d_i^2$ | $n_{\text{test}}^2 + \sum_{i=2}^d d_i^2 + \prod_{i=2}^d d_i$ |

Finally, time complexities when using Lanczos decompositions throughout are shown in Table 3, with the corresponding memory requirements after pre-computation shown in Table 4. These present further improvements to the Cholesky based approaches described throughout and enable Matheron's rule sampling with MTGPs to scale to larger $n$ and larger $t$ than even the exact settings.

Table 3: Time complexities for posterior sampling in single-output, multi-task, and high-order (HOGP) Gaussian Process models with LOVE fast predictive variances and Lanczos decompositions of rank $r$. Time complexities shown in blue are our contributions that have not yet been considered by the literature. Standard Sampling multi-task Gaussian processes scales multiplicatively in the combination of the number of tasks, $t$, and the number of data points, $n$, while using Matheron's rule allows for structure exploitation that reduces the combination to become additive in these components.

| Model | Distributional (Standard) (3) | With Matheron's rule (5) |
|---|---|---|
| Single-Output | $\mathcal{O}(r(n^2 + n_{\text{test}}^2))$ | $\mathcal{O}(r(n^2 + n_{\text{test}}^2))$ |
| Multi-Task | $\mathcal{O}(rt^2(n^2 + n_{\text{test}}^2))$ | $\mathcal{O}(r((n^2 + n_{\text{test}}^2) + t^2)$ |
| HOGP | — | $\mathcal{O}(r((n^2 + n_{\text{test}}^2) + \sum_{i=2}^d d_i^2))$ |

Table 4: Memory complexities after pre-computation for posterior sampling in single-output, multi-task, and high-order (HOGP) Gaussian Process models when using LOVE posterior covariances and Lanczos decompositions of rank $r$. Matheron's rule allows decomposition across the Kronecker product of the train and task covariances, enabling significant improvements in memory scaling.

| Model | Distributional (Standard) (3) | **With Matheron's rule** (5) |
|---|---|---|
| Single-Output | $n_{\text{test}}r$ | $n_{\text{test}}r + +nn_{\text{test}}$ |
| Multi-Task | $(n_{\text{test}}t)r$ | $r(n_{\text{test}} + t) + nt$ |
| HOGP | — | $r(n_{\text{test}} + \sum_{i=2}^d d_i) + \prod_{i=2}^d d_i$ |

### C.1.1 Extension to Linear Model of Coregionalization

We close this section by noting that the linear model of coregionalization (LMC) as described in Álvarez et al. [1] can be written as a sum of Kronecker products: $K_{\text{train}} = \sum_{q=1}^Q B_q \otimes K_q(X, X')$. We do not know of an efficient solve of sums of more than two Kronecker products, and we do not have a strong implementation of approximation methods or specialized preconditioners for solves of the form $(K_{\text{train}} + T_l)^{-1}z$. However, exploiting Kronecker strucutre, matrix vector products $K_{\text{train}}v$ cost $\mathcal{O}(Qnt(n+t))$ so that we can use conjugate gradients to compute solves and Lanczos to compute root decompositions in $\mathcal{O}(rQnt(n + t))$ time and $\mathcal{O}(rnt)$ memory. We can similarly compute a dense root decomposition update to form $MM^\top \approx \mathcal{K}_{\text{joint}}$ by following the same strategy as before (e.g. root updates [34]) but on matrices of size $nt$ and with an update of size $n_{\text{test}}t$. The structured MVMs make the updates more efficient, as computing $L_{22}$ costs only $rQn_{\text{test}}t(n_{\text{test}} + t)$ and $L_{12}$ costs $rQnt(n_{\text{test}} + t)$ to form the dense $r \times (n + n_{\text{test}})t$ updated root decomposition $M$. Thus, we can achieve efficient sampling using Matheron's rule and Lanczos variance estimates in effectively $\mathcal{O}(rQt(n^2 + n_{\text{test}}^2))$ time.

By comparison, sampling using the distributional approach would require dense factorizations of non-structured matrices, e.g. $\Sigma^*$, that do not have fast MVMs, thereby proving to be more expensive both computationally and memory wise. Indeed, the advantages will be magnified for large $n_{\text{test}}$ as then forming and decomposition $\Sigma^*$ may quickly become too expensive memory wise. We can exploit structured MVMs for sampling using Matheron's rule. Implementation wise, this provides $3x$ speedups on a single GPU, while significantly improving the memory overhead; however, we leave detailed exploration for future work.

### C.2 Autokrigeability and the HOGP Model

The High-Order Gaussian Process model has latent parameters, $x_l$, for each latent dimension, so that $K_i = k(x_i, x_i)$. Zhe et al. [64] initialize $x_l \sim \mathcal{N}(0, I)$ and optimize them as nuisance hyper-parameters, possibly regularizing them with weight decay. For completeness, they consider multi-dimensional latent dimensions, e.g. $x_l$ is a matrix, while we consider here $x_l$ as only one dimensional. Our analysis holds for multi-dimensional latents.

In the noiseless limit, the HOGP model falls prey to autokrigeability as described by Bonilla et al. [6]. If we were predicting a vector, this would not be an issue; however, we are predicting a tensor. In general, we expect the tensor's dimensions to be smoothly varying — that is, as we move down a row, we expect the covariance to be smoothly varying (e.g. it has smooth spatial structure). This prior assumption can be demonstrated on the variance as shown in Figure A.1a for simulated data: as we move down rows and columns, the sample variance stays at least somewhat consistently high (the first column) or low (the tenth column).

Considering $n = 1$ test point, only one latent dimension, and then taking the limit as $\sigma^2 \to 0$, the posterior variance becomes

$$\Sigma := K_{x_{\text{test}}x_{\text{test}}} \otimes K_L - (K_{x_{\text{test}}x} \otimes K_L)(K_{XX}^{-1} \otimes K_L^{-1})(K_{Xx_{\text{test}}} \otimes K_L)$$
$$= (K_{x_{\text{test}}x_{\text{test}}} - K_{x_{\text{test}}X}K_{XX}^{-1}K_{Xx_{\text{test}}}) \otimes K_L$$
$$= aK_L,$$

with $a = (K_{x_{\text{test}} x_{\text{test}}} - K_{x_{\text{test}} X} K_{XX}^{-1} K_{X x_{\text{test}}})$ (a scalar). The posterior variances for each output are given by the diagonals of $K_L$ or of the diagonal of $\otimes_{i=2}^{d} K_i$ for a $d$-tensor. The covariances between outputs are given by $K_L$.

The trouble arises from the fact that if the latent parameters, $v_l$, are not smoothly varying across $l$ then $K_L$ will not be smoothly varying either. A priori, we might expect that for a given set of outputs in the tensor, say indices $0, 1, 2$, that their posterior variances would also be smoothly varying (as the underlying process across the tensor is "smooth" in some sense), shown in Figure A.1a. Referring back to Figure A.8 for intuition, by smooth, we mean that each pixel in the outcome maps is close in some sense to its nearest eight neighbors. We should then expect the model's predictive posterior variances to vary in a similar fashion to the model's predictive posterior, which is data dependent.

For the HOGP tensor, the predictive posterior variance over an entire tensor (e.g. the coverage maps with 3 dimensions) is given by the product of the diagonal of each posterior. Thus, the inter-latent relationships can very quickly produce a "jagged" posterior variance as shown in Figure A.1c, with the posterior covariance becoming even more highly patterned (Figure A.1b).

For small $\sigma^2$, we can approximate $\Sigma$ as $aK_L$ with $a = (K_{x_{\text{test}} x_{\text{test}}} - K_{x_{\text{test}} X}(K_{XX}^{-1} + \sigma^2 K_{XX}^{-2}) K_{X x_{\text{test}}})$ and consider the properties of $K_L$. For smoothly varying covariances between indices in the posterior covariance matrix, we want smoothly varying $K_L$ and thus smoothly varying $x_l$.

**Smoothly Varying Latent Dimensions: the HOGP + GP** To produce smoothly varying latent dimensions, we initialize $x_l$ as a random draw from a multivariate normal distribution (or a Gaussian process) with zero mean and with a Matern 2.5 kernel and lengthscale 1 in all of our experiments. The kernel is evaluated on $(0, 1./d_i, 2./d_i, \cdots, (d_i - 1)/d_i, 1.)$ for inputs. We then use this distribution as a prior on the $x_l$ latents as well to help produce smoothly varying latents.

Example prior draws are shown in Figure A.1f in orange for two of its latent dimensions. The induced posterior variances are shown in Figure A.1e, which are considerably more smoothly varying than the random initializations. It is somewhat closer to the true variances of the function (these are un-trained models). Similarly, the (squeezed) posterior covariance matrix as shown in Figure A.1d shows much less covariance patterning than the random covariances in Figure A.1b. Importantly, after training the model, the largest impact is not on the predictive mean, but rather the posterior covariances and thus the posterior samples. We refer to HOGP models with this type of latent dimension prior and initialization as the HOGP + GP in the main text. We leave a theoretical analysis of these priors to future work. In Figure A.1, the true latent function is $f(x, y) = \sin(2x * i) * \cos(0.4yj) + \epsilon$, where $i, j$ are the tensor indices (here, $(0, 31)^2$) and $\epsilon \sim \mathcal{N}(0, 0.01^2)$.

### C.3 Convergence of Sample Average Approximation of Monte-Carlo Acquisition Functions

Following Balandat et al. [4], we consider the following class of acquisition functions:

$$\alpha(x; \Phi, y) = \mathbb{E}\big[a(g(f(x)), \Phi) \,|\, Y = y\big], \tag{A.4}$$

Here $x \in \mathbb{R}^{q \times d}$ is a set of $q$ candidate points, $g : \mathbb{R}^{n_{\text{test}} \times t} \to \mathbb{R}^{n_{\text{test}}}$ is an objective function, $\Phi \in \mathbf{\Phi}$ are parameters independent of $x_{\text{test}}$ in some set $\mathbf{\Phi}$, and $a : \mathbb{R}^{n_{\text{test}}} \times \mathbf{\Phi} \to \mathbb{R}$ is a utility function that defines the acquisition function.

Letting $\xi^i(x)$ denote a sample from $f(x)|(Y = y)$, we have the following Monte Carlo approximation of (A.4):

$$\hat{\alpha}_N(x; \Phi, y) := \frac{1}{N} \sum_{i=1}^{N} a(g(\xi^i(x)), \Phi) \tag{A.5}$$

Suppose $\mathbb{X} \subset \mathbb{R}^d$ is a feasible set (the "search space"). Let $x^{\text{opt}} := \arg\max_{x \in \mathbb{X}^{n_{\text{test}}}} \alpha(x; \Phi, y)$ and denote by $\mathcal{X}^{\text{opt}}$ the associated set of maximizers. Similarly, let $\hat{x}_N^{\text{opt}} := \arg\max_{x \in \mathbb{X}^{n_{\text{test}}}} \hat{\alpha}_N(x; \Phi, y)$. Then we have the following:

**Proposition 1.** *Suppose that $\mathbb{X}$ is compact, $f$ has a GP prior with continuously differentiable mean and covariance functions, and $g(\cdot)$ and $a(\cdot \Phi)$ are Lipschitz continuous. If the base samples $\{v^i\}_{i=1}^{n+n_{\text{test}}}$ and $\{\epsilon^i\}_{i=1}^{n}$ are i.i.d. with $v^i \sim \mathcal{N}(0, 1)$ and $\epsilon^i \sim \mathcal{N}(0, \sigma^2)$, respectively, then*

*1. $\hat{\alpha}_N^{\text{opt}} \to \alpha^{\text{opt}}$ a.s.*

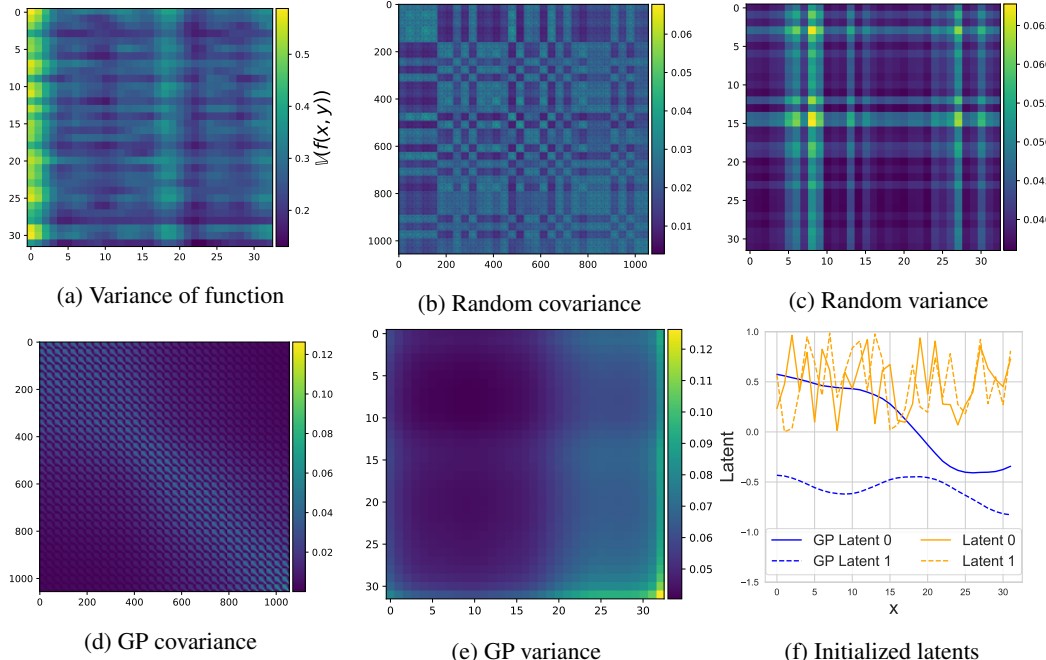

Figure A.1: **(a)** Sample variance of true function draws over the indices. **(b)** Posterior covariance from a HOGP model initialized with random latent dimensions shown as orange lines in **(f)**. **(c)** Posterior variance, shown on the output dimensions for random latents, the variance varies jaggedly. **(d)** Posterior covariance from a HOGP model initialized with latent dimensions drawn from a GP, shown as blue lines in **(f)**. **(e)** Posterior variance, shown on the output dimensions for the GP latents; the variance now varies smoothly. **(c,e)** Flattened (each output is now a single pixel) posterior covariance for random and GP-drawn latents. The GP-drawn latent covariance is much more smoothly varying.

2. $dist(\hat{x}_N^{opt}, \mathcal{X}^{opt}) \to 0$ *a.s.*

To prove Proposition 1, we need the following intermediate result:

**Lemma 1.** *Under Matheron sampling, $a(g(\xi(x)), \Phi) = a(g(h(x, \tilde{\epsilon})))$ with $h : \mathbb{R}^{n_{test} \times d} \times \mathbb{R}^{2n + n_{test}}$ and $\tilde{\epsilon} \in \mathbb{R}^{2n + n_{test}}$ a random variable. Moreover, there exists an integrable function $\ell : \mathbb{R}^{2n + n_{test}} \to \mathbb{R}$ such that for almost every $\tilde{\epsilon}$ and all $x, x' \in \mathbb{X}$,*

$$\left| a(g(h(x, \tilde{\epsilon}))) - a(g(h(x', \tilde{\epsilon}))) \right| \le \ell(\tilde{\epsilon}) \| x - x' \|. \tag{A.6}$$

*Proof of Lemma 1.* From (5) we have that under Matheron sampling a posterior sample is parameterized as

$$\xi(x) = f + K_{xX}(K_{XX} + \sigma^2 I)^{-1} y - (K_{XX} + \sigma^2 I)^{-1}(Y + \epsilon) \tag{A.7}$$

where $(f, Y)$ are joint samples from the GP prior. We parameterize $(f, Y)$ as $(f, Y) = R(x)\nu$ where $R(x)$ is a root[2] of the covariance from (6), and $\nu \sim \mathcal{N}(0, I)$. We can thus write (A.7) as

$$\xi(x) = \begin{bmatrix} I & -M(x) & 0 \\ 0 & 0 & M(x) \end{bmatrix} \begin{bmatrix} R(x)\nu \\ y - \epsilon^i \end{bmatrix} = M(x)y + A(x)\tilde{\epsilon} \tag{A.8}$$

where $M(x) := K_{xX}(K_{XX} + \sigma^2 I)^{-1}$,

$$A(x) := \begin{bmatrix} I & -M(x) & 0 \\ 0 & 0 & -\sigma M(x) \end{bmatrix} \begin{bmatrix} R(x) & 0 \\ 0 & I \end{bmatrix}$$

---

[2]For simplicity we assume that $R(x) \in \mathbb{R}^{n + n_{\text{test}}}$ for all $x$, but the results also apply to lower-rank roots (the argument follows from simple zero-padding.

and $\tilde{\epsilon}_j \sim \mathcal{N}(0,1)$ for $j = 1, \ldots, 2n + n_{\text{test}}$. Therefore,

$$\|\xi(x) - \xi(x')\| \leq \|M(x) - M(x')\|y + \|A(x) - A(x')\| \|\tilde{\epsilon}\|.$$

From the arguments of Balandat et al. [4], the assumption of continuously differentiable mean and covariance functions and the compactness of $\mathbb{X}$ imply that there exist $C_M, C_A < \infty$ s.t. $\|M(x) - M(x')\| \leq C_M$ and $\|A(x) - A(x')\| \leq C_A$ for all $x, x' \in \mathbb{X}$. Moreover, since both $g(\cdot)$ and $a(\cdot \Phi)$ are Lipschitz, there exists $L < \infty$ such that $|a(g(h(x, \tilde{\epsilon}))) - a(g(h(x', \tilde{\epsilon})))| \leq L\|\xi(x) - \xi(x')\|$. Consequently, (A.6) holds with $\ell(\tilde{\epsilon}) = LC_M y + LC_A \tilde{\epsilon}$, which is integrable since (i) $y$ is almost surely finite and $\tilde{\epsilon}_j \sim \mathcal{N}(0,1)$. $\square$

*Proof of Proposition 1.* Lemma 1 mirrors Lemma 1 from the supplementary material of Balandat et al. [4], and shows the corresponding result for the posterior samples parameterized under sampling using Matheron's rule. Proposition 1 then follows from the same arguments as in the proof of Theorem 1 in Balandat et al. [4]. $\square$

Similar to Balandat et al. [4], it is also possible to show the following:

**Proposition 2.** *If, in addition to the assumptions of Proposition 1, (i) for all $x \in \mathbb{X}^{n_{test}}$ the moment generating function $t \mapsto \mathbb{E}[e^{ta(g(h(x,\tilde{\epsilon})))}]$ is finite in an open neighborhood of $t = 0$, and (ii) the moment generating function $t \mapsto \mathbb{E}[e^{t\ell(\epsilon)}]$ is finite in an open neighborhood of $t = 0$, then $\forall \, \delta > 0$, $\exists \, K < \infty$, $\beta > 0$ s.t. $\mathbb{P}(\text{dist}(\hat{x}_N^{opt}, \mathcal{X}^{opt}) > \delta) \leq Ke^{-\beta N}$ for all $N \geq 1$.*

This follows from the proof of Proposition 1; we can use exactly the same argument as in the proof of Theorem 1 in Balandat et al. [4].

# D  Further Experiments and Experimental Details

In this Appendix, we give further experimental details as well as some more experiments for the applications of Matheron's rule to various Bayesian Optimization tasks.

Unless otherwise specified, all data is simulated. The code primarily relies on PyTorch [44] (MIT License), BoTorch [4] (MIT License), GPyTorch [28] (MIT License).

## D.1  Drawing Posterior Samples

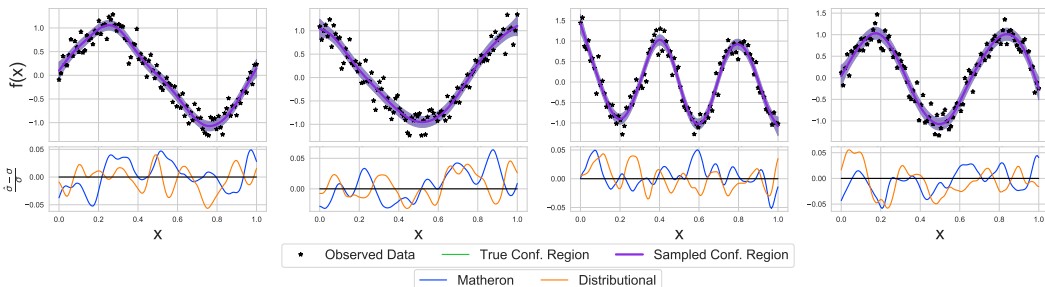

Figure A.2: **Top:** Posterior mean and confidence regions for a two-task GP. The Matheron's rule sampled confidence region is overlaid on top of the true confidence region; these are visually indistinguishable. **Bottom:** Relative error of the estimated standard deviation from $1024$ samples drawn from the predictive posterior using either Matheron's rule or distributional sampling; plotted as a function of $\hat{x}$. Again, the samples are effectively indistinguishable.

In Figure A.2, we show the accuracy of Matheron's rule sampling, where it is indistinguishable from conventional sampling from a MTGP (3) in terms of estimated standard deviations as well as the confidence regions. Here, we drew $1024$ samples from both sampling mechanisms and used the true mean and variance of the GP predictive posterior to shade the confidence regions. This result is to be expected as Matheron's rule draws from exactly the same distribution as the predictive posterior.

**Experimental Details** For the plots in Figure 2, we fit a Kronecker multi-task GP with data from the Hartmann-$5D$ function as in Feng et al. [26] and Appendix D.2 for 25 steps with Adam, before loading the state dict into the various implementations. We followed the same procedure for the LOVE experiments in Figure A.3.

The GPU experiments were run sequentially over 10 trials on a single $V100$ GPU with 16GB of memory. We show the mean over the ten trails and two standard errors of the mean (on a `p3.2xlarge` AWS instance). The CPU experiments were run sequentially over 6 trails on a single CPU and given 128GB of memory. Specifically, this corresponds to a `c5d.18xlarge` AWS instance.

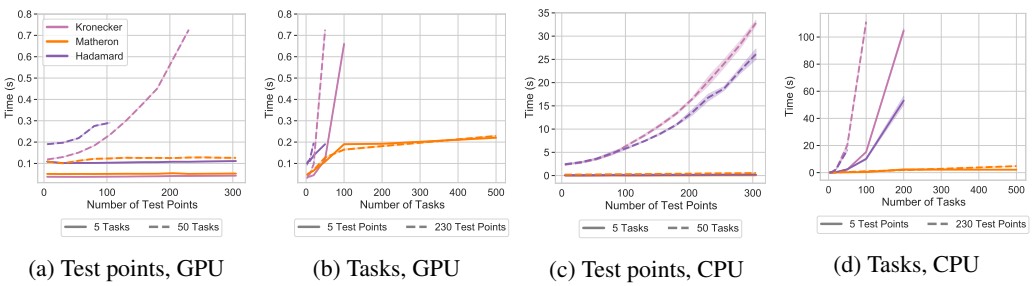

|  (a) Test points, GPU | (b) Tasks, GPU | (c) Test points, CPU | (d) Tasks, CPU |

Figure A.3: Timings for distributional sampling using LOVE cached predictive covariances with Hadamard (LCEM) and Kronecker (LCEM + Kronecker) variants of MTGPs as well as our sampling based on Matheron's rule (LCEM + Matheron) as the number of test points vary for fixed tasks **(a,c)** and as the number of tasks vary for fixed test points **(b,d)** on a single Tesla V100 GPU **(a,b)** and on a single CPU **(c,d)**. The multiplicative scaling of the number of tasks and data points creates significant timing and memory overhead, causing the Kronecker and Hadamard implementations to run out of memory for all but the smallest numbers of tasks and data points.

## D.2 Multi-Task BO and Contextual BO with LCE-M Contextual Kernel

We now consider contextual BO (CBO), an extension of multi-task BO, following Feng et al. [26] and using their embedding-based kernels, the LCE-M model. In CBO, the objective is to maximize some function over all observed contexts; given experimental conditions and a query, we want to choose the best action across all of the possible settings (the average in our experiments). Feng et al. [26] considered contextual BO (CBO), and proposed the LCE-M model, an MTGP using an embedding-based kernel. LCE-M models each context as a task; all contexts are observed for any given input, and the multi-task kernel is then $k_n(x, x')k_t(i, j) = k_n(x, x')k_t(E(c), E(c'))$, where $E$ is a nonlinear embedding and $c$ are the contexts.

We consider both multi-task and contextual versions of this problem. For the multi-task setting, we do not actually perform contextual optimization (selecting a different candidate for each context), but find a single action that maximizes the average outcome across all observed contexts. For both settings, we would intuitively expect that using context-level observations can improve modelling and thus optimization performance. We use qEI (batch expected improvement with MC acquisition, $q = 2$, Balandat et al. [4]) on the objective and compare the (Hadamard) LCE-M model, a Kronecker variant, and one based on Matheron MTGP sampling.

**Multi-task Bayesian Optimization:** Figure A.4 shows results on the multi-task version of this problem for $5, 10, 20, 50, 100$ contexts. We can see that, as expected, the optimization performance is identical for the three sampling methods on five contexts; the $50$ and $100$ context case shows that our Matheron-based sampling achieves better performance overall as the other methods run out of memory. In Figure A.4f, we also show the average and steps achieved (confidence bars are two standard errors of the mean) as the number of tasks (contexts) increases. This clearly shows the impact of the increased memory usage for both the LCEM and LCEM + Kronecker implementations, as they run out of memory after only a few steps, rather than being able to reach all $150$ optimization steps.

**Contextual Bayesian Optimization:** In Figure A.5, we display the results of setting up the Hartmann-$5D$ problem as a contextual Bayesian Optimization. Here, we observe each context

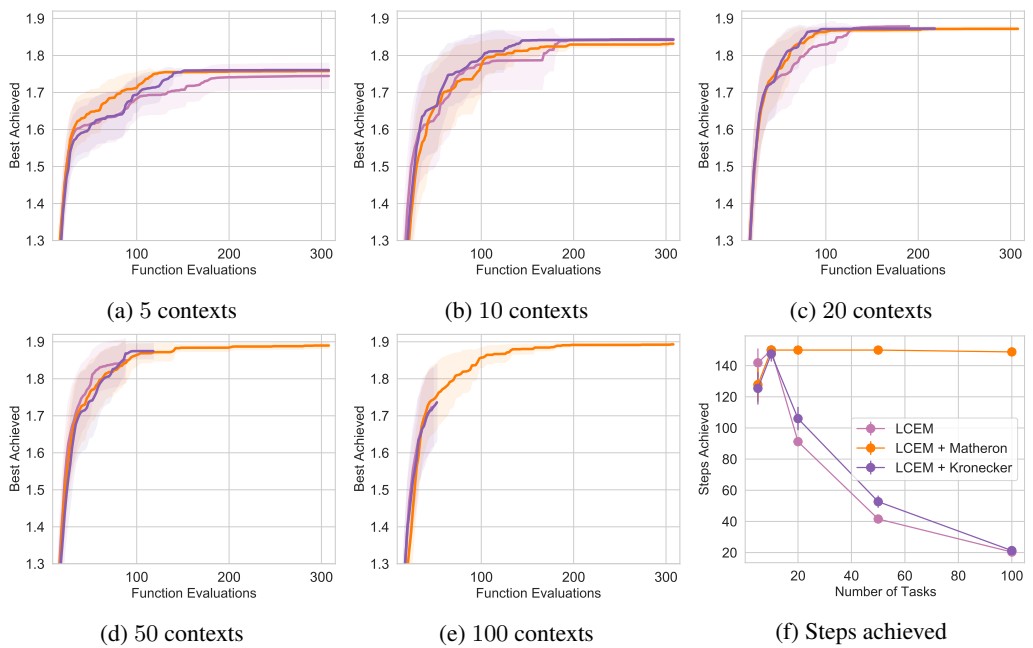

Figure A.4: Multi-task Bayesian Optimization with on Hartmann5D with LCE-M kernel using distributional sampling with Hadamard (LCEM) and Kronecker (LCEM + Kronecker) models, and our Matheron-based posterior sampling (LCE-M + Matheron). For five contexts **(a)**, optimization performance is essentially identical as expected; for 100 contexts **(b)**, only the LCEM + Matheron's rule model reaches a maximum as the others run out of memory. Results over 40 trials for 10, 20, and 50 contexts on Hartmann-6 translated into a contextual problem. We also show the number of average steps achieved in **(d)** where only LCEM + Matheron's rule is able to complete all 150 steps.

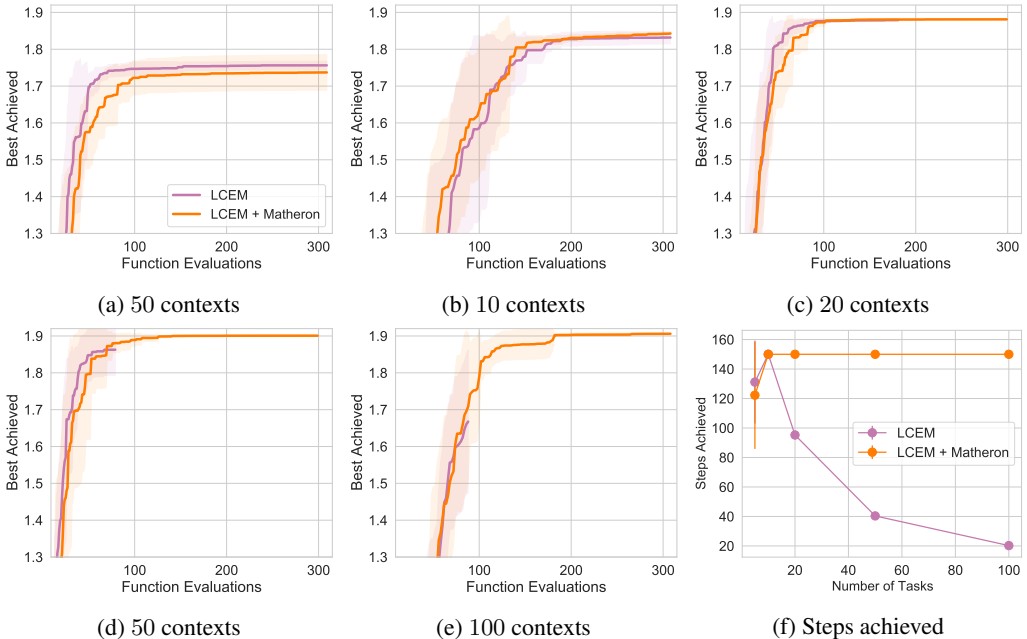

Figure A.5: Results over 8 trials for 10, 20, and 50 contexts on Hartmann-6 translated into a contextual problem. We also show the number of average steps achieved in **(d)** where only LCEM + Matheron's rule is able to average 150 steps (the maximum that we ran each trial for) of Bayesian Optimization.

at every iteration, but need to choose a policy to for a randomly chosen context. We then observe all contexts' observations for that observed policy, and plot the best average reward achieved across all contexts. Again, the findings from the MTGP experiment carry over — which is that optimization performance is nearly identical, but that the Matheron's rule implementation can achieve many more steps at a higher number of contexts, as shown in Figure A.5f.

**Experimental Details:** We fit the MTGPs with a constant diagonal likelihood with a Gamma$(1.1, 0.05)$ prior on the noise, ARD Matern kernels on the data with $\nu = 2.5$, lengthscale priors of Gamma$(3.0, 6.0)$, and a Gamma$(2.0, 0.15)$ prior on the output dimension using Adam for 250 steps. The LCEM kernel uses similar priors and a one dimensional embedding, following Feng et al. [26]. The LCEM implementation was Hadamard based and exactly from `https://github.com/pytorch/botorch/blob/master/botorch/models/contextual_multioutput.py`. For all, we used 10 initial points, ran 150 steps of BO, normalized the inputs to $[0, 1]^d$ and standardized the responses. The simulated Hartmann-$5D$ function comes from `https://github.com/facebookresearch/ContextualBO/` (MIT License) [26]. Here, we use a single 16GB V100 GPU (p3.8xlarge AWS instance) and repeat the experiments 43 times for 5 tasks and 40 times for 100 tasks, showing the mean and two standard errors of the mean. For BO loops, we used 128 MC samples, $q = 2$ batch size, 256 initialization samples with a batch limit of 5, 10 restarts for the optimization loop and 200 iterations of L-BFGS-B. All other options were botorch defaults including the LKJ prior with $\eta = 2.0$ over the inter-task covariance matrix [40].

### D.3 Constrained Multi-Objective Bayesian Optimization

For C2DTLZ2, we initialized with 25 random points, while with OSY we initialized with 14 random points, chosen via rejection sampling to find at least 7 feasible points. For both, we then optimized with 128 MC samples, 10 random restarts, 512 base samples, a batch limit of 5, an initialization batch limit of 20 and for up to 200 iterations of L-BFGS-B. We used a batch size of $q = 2$ for C2DTLZ2 optimizing for 200 steps and a batch size of $q = 10$ for OSY, optimizing for 30 steps. We used 16 random seeds for OSY and 24 for C2DTLZ2 and plot the mean and two standard errors of the mean. For both, we used the default reference points as $(1.1, 1.1)$ and $(-75, 75)$ for the EHVI approaches.

The C2DTLZ2 function comes from BoTorch, while the OSY function is a reimplementation of `https://github.com/msu-coinlab/pymoo/blob/master/pymoo/problems/multi/osy.py` (Apache 2.0 License). For these, we used 24GB Nvidia RTX GPUs on an internal server.

### D.4 Scalable Constrained Bayesian Optimization

In Figure A.6, we show the wall clock time for fitting the batched GPs and the MTGPs on the lunar lander problem with $m = 50$ constraints. Here, the MTGPs are faster because they are somewhat more memory efficient; note that several runs for both reached convergence during optimization very quickly after reaching BoTorch's default Lanczos threshold ($n = 800$) thus decaying the model fitting times. We also show the time required to sample all tasks, again finding that the Thompson sampling time is much slower for the batched GPs. Shown are means and log normal confidence intervals around the mean (hence the asymmetry). In Table 5, we show the number of steps and the proportion of succeeded trials required to reach feasibility, finding that the MTGPs also reduced the number of steps required to achieve feasibility and thus improved the number of feasible runs. For steps to feasibility, we again show means and log normal CIs around the mean.

Here, we followed the parameterizations and other implementation details from Eriksson and Poloczek [25] and used 30 random seeds for the lunar lander problems and 9 on the MOPTA08 problem (8 for batch GPs due to memory issues). Here, we used a single 24GB Titan RTX GPU for all experiments (part of an internal server), and used KeOPS [27] for the batched GPs. We used a full rank ICM kernel with a LKJ prior $\eta = 2.0$ [40] and a smoothed box prior on the standard deviation of $(e^{-6}, e^{1.25})$. for the multi-task GPs and diagonal Gaussian noise with a Horseshoe$(0.1)$ prior, constraining the diagonal noise to $[10^{-6}, 4.0]$.

The executable for MOPTA08 is available at `https://www.miguelanjos.com/jones-benchmark` (no license provided). The lunar lander problem uses `https://github.com/openai/gym/blob/master/gym/envs/box2d/lunar_lander.py` [8] (MIT

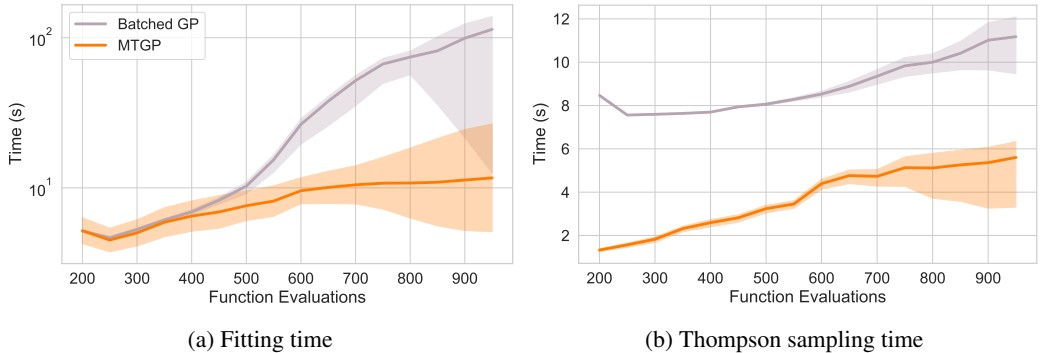

|  | (a) Fitting time | (b) Thompson sampling time |

Figure A.6: **(a)** Wall clock time for model fitting with multi-task and batched Gaussian processes as a function of the number of function queries. **(b)** Time for Thompson sampling as a function of the number of function queries. In both cases, the multi-task Gaussian process is faster; training due to using conjugate gradients and Kronecker MVMs while Thompson sampling is faster due to the Matheron's rule approach that we use.

Table 5: Number of function evaluations to achieve feasibility on the lunar lander (LL) and the MOPTA08 optimization problems, given that feasibility was reached, as well as the proportion of runs that achieved feasibility. Using a multi-task GP in SCBO achieves feasible outcomes with fewer function evaluations; on the $m = 100$ constraint lunar lander, the batch GPs ran out of memory before reaching feasibility.

| Method | Time to Feasibility | Proportion |
|---|---|---|
|  | LL, $m = 50$ |  |
| MTGP SCBO | $314, (229, 333)$ | $26/30, (0.73, 1.0)$ |
| Batch SCBO | $401, (273, 432)$ | $24/30, (0.64, 0.96)$ |
| PCA GP SCBO | $324, (239, 348)$ | $25/30, (0.68, 0.97)$ |
|  | LL, $m = 100$ |  |
| MTGP SCBO | $349, (260, 390)$ | $17/30, (0.33, 0.8)$ |
| Batch SCBO | — | $0/30$ |
|  | MOPTA08 |  |
| MTGP SCBO | $292, (250, 327)$ | $9/9$ |
| Batch SCBO | $415, (318, 493)$ | $8/8$ |

License). The SCBO code from Eriksson and Poloczek [25] is currently unreleased; we implemented our own version.

### D.5   Composite Bayesian Optimization Experiments

In all experiments with the HOGP, we used diagonal Gaussian likelihoods with Gamma$(1.1, 0.05)$ priors on the standard deviation, and Matern 2.5 kernels with a lengthscale prior of Gamma$(3., 6.)$. For the standard version, we randomly initialized latent parameters to be standard normal, while for the HOGP + GP models, we randomly sampled the latents from a Matern 2.5 kernel with lengthscale 1 and input values as the indices, using the kernel as the covariance for a zero mean multivariate normal prior. For all experiments we used qEI with a batch size of 1.

For the environmental problem, we followed the implementations of Balandat et al. [4], Astudillo and Frazier [3], and used 8 random restarts, 256 MC samples, and 512 base samples, a batch limit of 4, and an initialization batch limit of 8. These experiments were performed 50 times on 16GB V100 GPUs (part of an internal cluster). The bounds are $(7, 0.02, 0.01, 30.01)$ and $(13.0, 0.12, 3., 30.295)$.

For the PDE problem, we followed the example implementation given at `https://py-pde.readthedocs.io/en/latest/examples_gallery/pde_brusselator_expression.html#sphx-glr-examples-gallery-pde-brusselator-expression-py` (MIT License). For the

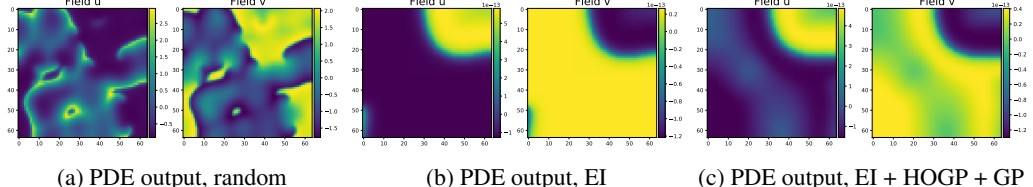

| (a) PDE output, random | (b) PDE output, EI | (c) PDE output, EI + HOGP + GP |

Figure A.7: **(a)** Example solution from the Brusselator problem. **(b)** PDE solution as found via EI on the metric itself. **(c)** PDE solution as found via composite BO using EI with the HOGP model, which displays less variance than the EI solution. Running BO to minimize the variance is able to easily find settings of parameters within the bounds that push the reactivity to zero; the variance is reduced overall by using the HOGP + GP model. All solutions are de-meaned.

metrics, we computed the weighted variance and minimized that function. Non-finite outputs were set to $1e5$. We up-weighted the weights on the first two rows and columns for each output to have weights $10x$ that of the rest of the inputs. These were run 20 times with 5 initial points, 50 optimization steps, using 64 MC samples and 128 raw samples with 4 optimization restarts. These were run on CPUs with 64GB of memory (part of an internal cluster). An example output, as well as solutions found by EI (objective value 0.1088) and composite EI using the HOGP model (objective value 0.0087) are shown in Figure A.7.

On the radio frequency coverage problem, we initialized with 20 points, downsampled the two $241 \times 241$ outputs to $50 \times 50$ for simplicity, ran the experiments over 20 random seeds and for 150 steps. We used 32 MC samples, 64 raw samples with a batch limit of 4 and an initialization batch limit of 16. These were run on either $V100s$ with 32GB of memory or $RTX8000s$ with 48GB of memory on a shared computing cluster so we cannot tell which one was used. The problem is 30 dimensional and the first 15 dimensions are $(0.0, 10)^{15}$ with the second 15 coming as $(30.0, 50.0)^{15}$. The first dimensions correspond to the downtilt angles of the transmitters (in $3D$ coordinates), while the second set corresponds to the power levels of the transmitters.

For metric specfication, we used the following equations, following Dreifuerst et al. [21], where $R$ is the first output and $I$ is the second output:

$$Cov_{\text{f, strong}} = \sum_{i,j}^{50} \text{sigmoid}(-80 - R)$$

$$Cov_{\text{g, weak, area}} = \text{sigmoid}(R + 80) * \text{sigmoid}(I + 6 - R)$$

$$Cov_{\text{g, weak}} = \sum_{i,j}^{50} \text{sigmoid}(I * Cov_{\text{g, weak, area}} + 6 - R * Cov_{\text{g, weak, area}})$$

$$Obj = 0.25 * Cov_{\text{f, strong}} + (1 - 0.75) * Cov_{\text{g, weak}}$$

using the final line as the objective to maximize. $-80$ is the weak coverage threshold, while 6 is the strong coverage threshold. Representative coverage maps are shown in Figure A.8, along side the maps of weak coverage and strong coverage, analogous to that of a random set of parameters in Figure 1.

This code was provided to us on request by the authors of Dreifuerst et al. [21].

On the optics problem, we used the simulator of Sorokin et al. [53], initialized with 20 samples and ran for 115 steps with 64 MC samples, 64 raw samples for initialization with a batch limit of 1. We used the same computing infrastructure as on the coverage problem above. to convert the problem of optimizing visibility into a BO problem rather than a reinforcement learning one, we reset the simulator to $(1e-4, 1e-4, 1e-4, 1e-4)$ each time we queried the problem and optimized the log visibility.

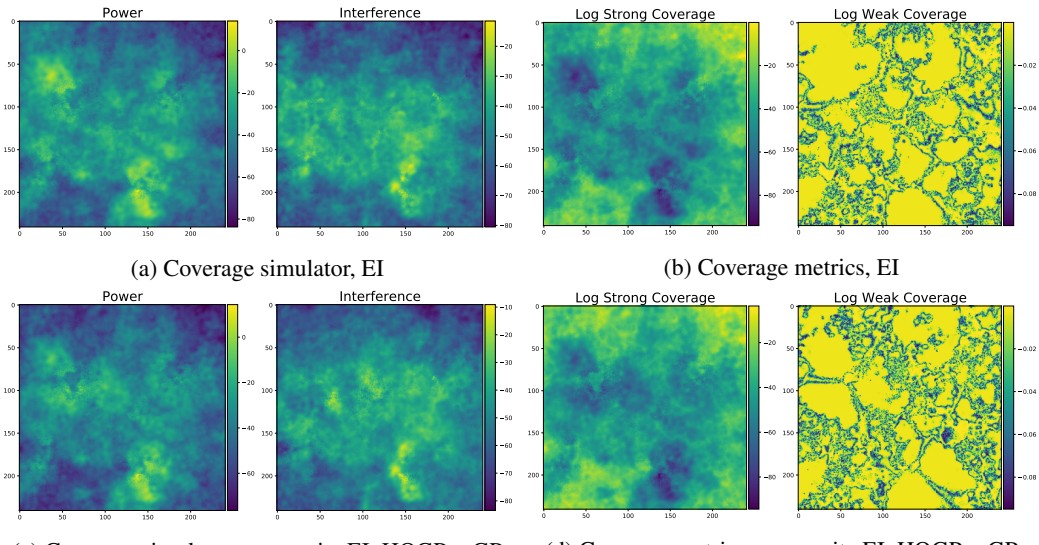

(a) Coverage simulator, EI       (b) Coverage metrics, EI

(c) Coverage simulator, composite EI, HOGP + GP      (d) Coverage metrics, composite EI, HOGP + GP

Figure A.8: **(a)** Coverage map obtained by optimizing EI on the aggregate metric, which yields the weak and strong coverage metrics (shown on log scale) **(b)**. **(c)** Coverage map obtained by optimizing EI using a composite objective with HOGP+GP, which yields the weak and strong coverage metrics (log scale) **(d)**. Composite BO with the HOGP yields distinctive differences between the best patterns found on the coverage metrics.

To increase signal, we up-weighted the center of the image as

$$\text{Intensity}_t := \sum_{i,j} \exp\{-(i/64 - 0.5)^2 - (j/64 - 0.5)^2\} * I_t$$

$$I_{\max} = \text{LogSumExp}(\text{Intensity}_t)$$

$$I_{\min} = -\text{LogSumExp}(-\text{Intensity}_t)$$

$$V = (I_{\max} - I_{\min})/(I_{\max} + I_{\min})$$

and maximized the logarithm of the visibility (V), where $I_t$ is the $t$th output of the model (there are 16 outputs, each is of shape $64 \times 64$).

We show several results from a single run in Figure A.9, where we see that only EI on the HOGP is able to at least partially align the two sets of mirrors; a random solution and EI on the metric keep the light coming from the two mirrors apart. The simulator itself comes from `https://github.com/dmitrySorokin/interferobotProject` (MIT License).

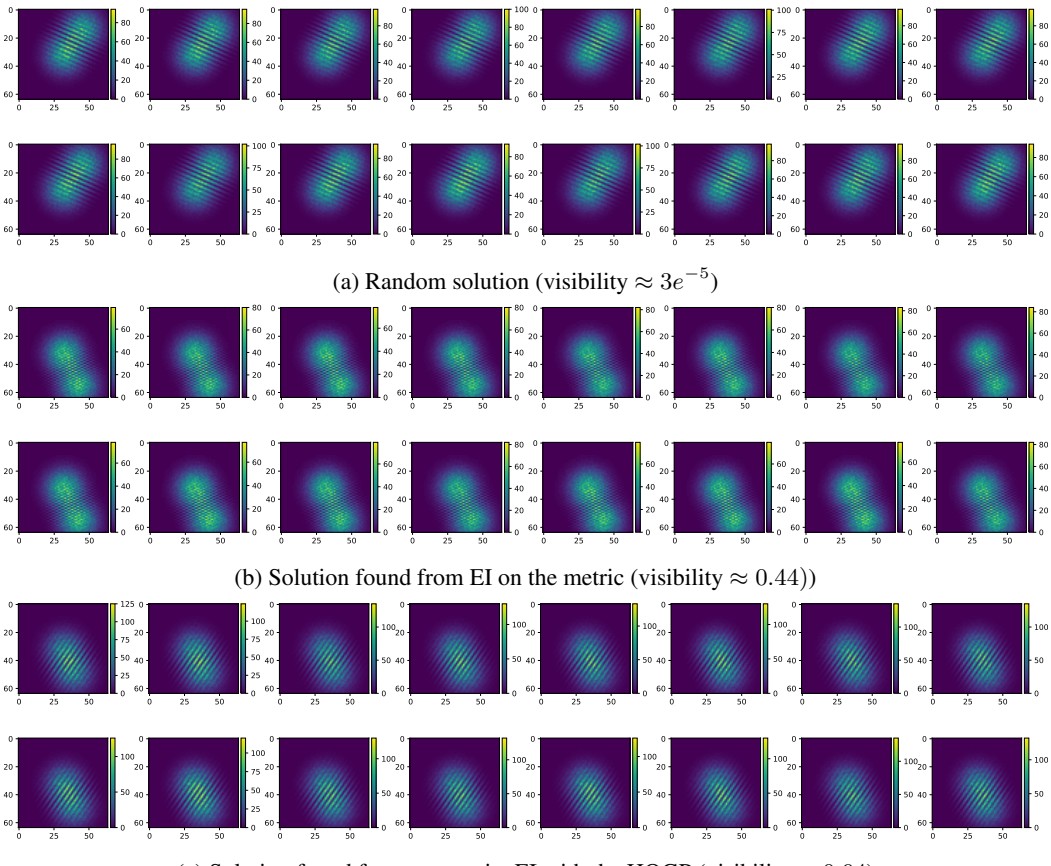

(a) Random solution (visibility $\approx 3e^{-5}$)

(b) Solution found from EI on the metric (visibility $\approx 0.44$))

(c) Solution found from composite EI with the HOGP (visibility $\approx 0.94$)

Figure A.9: Example simulator outputs for the optics problem. **(a)** a random movement produces very unaligned lights, while **(b)** EI on the metric itself somewhat aligns the two light sources. **(c)** Running composite BO with the HOGP model produces much more aligned light sources that are presented as much brighter on the scales.