# OpenReview forum: "Bayesian Optimization with High-Dimensional Outputs"
_NeurIPS.cc/2021/Conference — NeurIPS 2021 Poster_

### Official Review · Reviewer_Dpyw · 2021-07-07

**Rating:** 7
**Confidence:** 5

**Summary:**

The authors propose to scale Gaussian process modeling and sampling to high-dimensional outputs with an intrinsic coregionalization model (ICM). They do it by leveraging both a Kronecker structure on the outputs (all tasks observed everywhere) and the so-called Matheron rule to update prior samples. High-order GP models are considered as well. A variety of application examples are presented to illustrate the benefits of the method and pushing the state of the art in terms of number of outputs when MC acquisition functions are used.

**Limitations And Societal Impact:**

Yes

**Main Review:**

The paper reads well but the methodological part is too short. Crucial points are left in the supplementary while this material raises some concerns and needs proof-reading.

Consequently, the main problem --in my opinion-- is that I don’t see how to implement the method and reproduce the results without resorting to the authors code. Actual Kronecker solves, determinant expressions are not given (just a reference to [41] in supplementary). Only statements about complexity are provided (e.g., L221).  Then several cases are described, e.g., input-varying noise (or correlation in the noise), that is, not \sigma^2 I but generic T_l as discussed in C.1, while it is unclear which ones can really be done (**). I think this can be solved by providing a detailed (complete) pseudo-code of the approach that should be in the main text (the one for HOGPs can be in supplementary). If space is needed, some of the applications can be entirely moved to supplementary as they are direct applications of existing MC acquisition functions.

The limitations discussed in supplementary A should be included in the main text.

(**) In fact, from C.1.1: some of these options are more like ideas for future research than actual results:  “We do not know of an efficient solve of sums of more than two Kronecker products, and we do not have a strong implementation of approximation methods or specialized preconditioners for solves of the form (K train + T_l ) −1 z” . It is contradicting previous statements. Either really show that it works or leave it as a perspective (the supplemental is already long enough).




Typos :
- B.1: there are i indices in sums but that’s it.
- L729: strucutre


### Post-rebuttal comments ###
I would like to thank the authors for their detailed response, which I think should be fully integrated in the paper and its supplementary to be more self contained. In particular, I appreciate the addition of the Matheron's rule Posterior Sampling for MTGPs algorithm (requiring expressions that where not referenced or given before). Assuming that all the changes proposed on the reviews will be made (perhaps adding another round of proof reading on the inevitably numerous notations), I am happy to increase my score.

**Time Spent Reviewing:**

6

---

> ### Author Response · Authors · 2021-08-10
> **Thanks for the feedback**
>
> Thanks for the review and comments. We would like to clarify a few points here.
>
> **Kronecker Algebra and Multi-task Gaussian Processes**
>
> While a good reference to using Kronecker algebra in the ML community is found in [Saatci, ‘12], it was first applied in the context of MTGPs in the ML community dating back to [Rougier, ‘08] and [Bonilla, et al ‘07], despite likely being known to some extent in the geostatistics community before that. For an explicit description of Kronecker matrix vector multiplication, please see Golub and van Loan, Matrix Computations, Section 1.3.7-1.3.8 and Section 12.3.
>
> More specifically, our MTGP formulation is a standard MTGP with an ICM kernel, which has tutorial implementations in both GPy (https://nbviewer.ipython.org/github/SheffieldML/notebook/blob/master/GPy/coregionalized_regression_tutorial.ipynb) and GPyTorch (https://docs.gpytorch.ai/en/stable/examples/03_Multitask_Exact_GPs/Multitask_GP_Regression.html) (two of the most popular GP packages available today); to the best of our knowledge, these implementations exploit Kronecker structure during training. The core difference between those implementations and ours (which is an extension of the GPyTorch implementation) is that we sample from the posterior distribution in a rather different and significantly more efficient manner --- using Matheron’s rule instead of distributional sampling.
>
> ***Kronecker Algebra***
>
> Matrix vector multiplies (MVMs) are efficient and can be computed from:
> \begin{align}
> 	z = (K_1 \otimes K_2)\text{vec}(A) = \text{vec}(K_2 A K_1^\top);
> \end{align}
> if $K_1 \in \mathbb{R}^{n_1 \times n_1}$ and $K_2 \in \mathbb{R}^{n_2 \times n_2}.$ As a result, computing $z$ costs $\mathcal{O}(n_1^2 + n_2^2 + n_1 n_2 (n_1 + n_2))$ time (see Section 12.3 of Golub and van Loan [corrected]).
> There are several other useful Kronecker product properties (again summarized from Saatchi, '12, Golub and van Loan, amongst other sources):
> $$
> (A \otimes B)(C \otimes D) = AC \otimes BD,
> $$
> if the shapes match,
> $$
> (K_1 \otimes K_2)^{-1} = (K_1^{-1} \otimes K_2^{-1}),
> $$
> if both matrices are invertible,
> and
> $$
> \log |K_1 \otimes K_2| = n_2 \log |K_1| + n_1 \log |K_2|
> $$
> Using these properties, we then are able to compute matrix inverses of Kronecker plus *constant* diagonal matrics as
> $$
> (K_1 \otimes K_2 + \sigma^2 I)^{-1} = (Q_1 \otimes Q_2)(\Lambda_1 \otimes \Lambda_2 + \sigma^2 I)^{-1}(Q_1 \otimes Q_2)^\top,
> $$
> where $K_i = Q_i \Lambda_i Q_i^\top$ (the eigen-decomposition of $K_i$).
>
> ***Structured Noise Terms***
>
> In general, there is no efficient way to compute matrix inverses of the form: $(K_1 \otimes K_2 + T_l)^{-1}$ where $T_l$ is a diagonal matrix that is non-constant. This is because we cannot separate out the eigenvalues of $T_l$ like we can for the constant diagonal case. One can use conjugate gradients to compute solves in that setting.
>
> However, as Rakitsch et al, '13 demonstrate, if we assume a structured noise term, e.g. a likelihood that is $\text{vec}(Y) \sim \mathcal{N}(f, \Sigma_X \otimes \Sigma_T),$ then there is an efficient method of computing matrix inverses and solves:
> $$
> (K_1 \otimes K_2 + \Sigma_X \otimes \Sigma_T)^{-1} = (Q_X \Lambda_X^{1/2} \otimes Q_T \Lambda_T^{1/2})(\tilde Q_1 \otimes \tilde Q_2)(\tilde \Lambda_1 \otimes \tilde \Lambda_2 + I)^{-1}(\tilde Q_1 \otimes \tilde Q_2)^\top (Q_X \Lambda_X^{1/2} \otimes Q_T \Lambda_T^{1/2})^\top ,
> $$
> where
> $\tilde Q_1 \tilde \Lambda_1 \tilde Q_1^\top = \Lambda_X^{-1/2}Q_X^\top K_1 Q_X \Lambda_X^{-1/2}$
> and $\tilde Q_2$ is similar. Rakitsch et al, '13 argue that this type of noise covariance allows both estimation of "residual correlation between tasks due to latent causes" as well as input dependent noise.
>
> We note that in this setting, we require two sets of eigen-decompositions, of $K_1$ and $K_2$ as well as $\Sigma_X$ and $\Sigma_T.$ This method also works if $\Sigma_X$ and/or $\Sigma_T$ is not diagonal, but is most useful if they are.
>
>
> ***Marginal Log-Likelihood Computation***
>
>
> Using the above identities, we can plug these into the standard GP log marginal likelihood, following Rakitsch et al, '13 and Stegle et al, '11.
> In the single task setting, we have:
> $\log p(y) = \frac{1}{2}\left(N \log 2\pi - \log |K+\sigma^2 I| - y^\top (K + \sigma^2 I)^{-1} y\right)$ (Eq. 5.8 in Rasmussen \& Williams '06).
>
> This is naturally extended into the multi-task setting (we show only the constant diagonal case but please see Rakitsch et al, '13 for the structured noise case as it is just plugging in the above inverses):
>
> $$
> \log p(Y) = \frac{1}{2}\left(NT \log 2\pi - \log |K_X \otimes K_T +\sigma^2 I| - \text{vec}(Y)^\top (K_X \otimes K_T + \sigma^2 I)^{-1} \text{vec}(Y)\right)
> $$
> The log determinant term simplifies to
> $$
> \log |K_X \otimes K_T +\sigma^2 I| = \log |\Lambda_X \otimes \Lambda_T + \sigma^2 I|
> $$
> which is just the determinant of a diagonal matrix.
> The quadratic form similarly simplifies (as described above)
> $$
> \text{vec}(Y)^\top(Q_X \otimes Q_T)(\Lambda_X \otimes \Lambda_T + \sigma^2 I)^{-1}(Q_X \otimes Q_T)^\top \text{vec}(Y)^\top.
> $$
>
> We then estimate the kernel hyper-parameters with gradient based optimization of the log marginal likelihood, following Stegle et al, '11 and Rakitsch et al, '13.
>
> The predictive means and variances of the MTGP are given by Bonilla et al, '07 via Kronecker algebra.
>
> **Extension to Linear Model of Co-regionalization**
>
> Indeed, this is a perspective for future research, which we can drop in the camera ready. However, the approach described does give significant computational gains (between 2-4x for a fixed number of samples) and we implemented a version for the rebuttal. We drew 150 training data points uniformly [0,1] and attempted to draw 64 posterior samples from the posterior distribution, using Q  = T linear components (that is, the number of components was equal to the number of tasks). We varied both the test points and number of tasks. The results are presented in the tables below. For the fixed tasks setting, we considered a constant $64$ test data points, and for the fixed test point settings we considered $T = 15$ tasks. Note that the dashes indicate out of memory errors. This experiment was performed on an AWS p3.2xlarge instance with a V100 GPU.
>
> | Tasks | Matheron | Distributional |
> |-------|----------|----------------|
> | 5     | 0.39     | 0.08           |
> | 10    | 1.17     | 0.51           |
> | 15    | 3.25     | 4.25           |
> | 20    | 4.22     | 6.55           |
> | 35    | 6.91     | 20.35          |
> | 50    | 9.79     | ---            |
> | 75    | 14.73    | ---            |
> | 100   | 19.94    | ---            |
>
> | Test Data Points | Matheron | Distributional |
> |------------------|----------|----------------|
> | 8                | 2.53     | 2.06           |
> | 16               | 2.57     | 2.07           |
> | 32               | 2.63     | 2.11           |
> | 64               | 3.15     | 4.34           |
> | 128              | 3.49     | 4.59           |
> | 256              | 3.15     | 5.27           |
> | 512              | 3.46     | 6.62           |
> | 1024             | 4.67     | 8.94           |
>
>
> **Algorithm Description**
>
> Indeed, this is a good idea which we can add; however, we are a little confused as to whether you are referring specifically to the Kronecker formulation and model training (which we describe above) or more generally for the high-level Bayesian Optimization algorithm; we try to respond to both below.
>
> ***Bayesian Optimization Loop***
>
> Here is an algorithm description for the Bayesian optimization procedure. In broad steps, it can be described as, given initial data $(X, Y)$ and a model:
>
> Repeat until maximum number of trials reached:
> - Fit MTGP using Type-II MLE estimation (e.g. maximizing the log marginal likelihood as described above) on data $(X, Y)$.
> - Use Matheron’s rule posterior sampling within gradient based optimization of an acquisition function (e.g. Eq. A.5) to choose new candidate points $X_{cand}$
> - Query expensive function, $f$, with inputs $X_{cand}$ to return $Y_{cand}$
> - Stack $X = (X, X_{cand})$ and $Y = (Y, Y_{cand})$.
>
> ***Matheron's rule Posterior Sampling for MTGPs***
>
> As a sub-algorithm, Matheron’s rule posterior sampling can be accomplished in the following manner:
>
> We assume knowledge of fitted GP model and new data points, $x_{\text{test}}.$
> - Compute $K_{X, X_{\text{test}}}$, $K_{x_{\text{test}}, x_{\text{test}}}$ and use these to compute $\mathcal{K}_{\text{joint}},$ defined in Eq. 7, via lines 683 - 686, apologies for not numbering these equations.
> - Draw samples $(f,Y)$ using Eq. A.3 as well as \epsilon (line 692).
> - Compute the solve, w, in Eq. 5 / A.2 (specifically mentioned in line 695) via an eigen-decomposition (see equations referenced above).
> - Compute $z = K_{x_{\text{test}} X} w$ via a Kronecker matrix vector product (equations referenced above).
> - Finally compute the addition to find $\bar{f} = f + z$ which is our random samples.
>
> **Conclusion**
> Thanks for catching the typos as well. We hope that the above detail and clarifications will satisfy your concerns.

---

> > ### Author Response · Authors · 2021-08-24
> > **Thanks for considering our comments**
> >
> > Hi reviewer Dpyw,
> >
> > We appreciate you taking the time to consider our comments on your review, and thank you for adjusting your score.
> >
> > One additional comment we'd like to make is that that the LMC results from C.1.1, while not essential to the key contributions of the paper, also speak to the generality of our approach and would benefit the community. For completeness, we also include the algorithm for sampling from the LMC below
> >
> > We assume knowledge of fitted GP model and new data points, $x_{\text{test}}.$
> > - Compute $K_{X, X_{\text{test}}}$, $K_{x_{\text{test}}, x_{\text{test}}}$
> > - Compute $K_{\text{joint}}$ and its dense root decomposition $LL^\top \approx K_{\text{joint}}$ defined in Eq. 7, via lines 683 - 686, following the approach outlined in Appendix E of Jiang et al, ‘20. Note that we now need to use a dense decomposition, rather than the Kronecker exploiting updates for the ICM kernel.
> > - Draw samples $(f,Y)$ using $L^\top z$ as well as $\epsilon$ (line 692).
> > - Compute the solve, w, in Eq. 5 / A.2 (specifically mentioned in line 695) via conjugate gradients (as implemented in Gardner et al, ‘18).
> > - Compute $z = K_{x_{\text{test}} X} w$ via a sum of Kronecker matrix vector products (equations referenced above).
> > - Finally compute the addition to find $\bar{f} = f + z,$ producing our random samples.

---

### Official Review · Reviewer_zeBf · 2021-07-11

**Rating:** 7
**Confidence:** 3

**Summary:**

This paper presents the idea of scalable multi-task Gaussian processes using a trick known as Matheron's rule. Efficient sampling from multi-task GPs is essential in solving many black box optimization problems with high-dimensional outputs including multi-objective constrained BO and composite functions. Matheron's rule is used to reduce the dependency of the number of tasks from a multiplicative factor to an additive factor. Experimental results are presented validating the benefits of the approach wrt. the computational resource requirements and also sample complexity.

**Limitations And Societal Impact:**

Limitations and societal aspects have been adequately discussed in the appendix.

**Main Review:**

**Strengths:**
- The problem studied in this paper is quite significant to the BO community. BO with high dimensional outputs is a recurring problem in many applications. Traditional ways of modeling multi-task GPs do not scale well - this paper presents a scalable approach to this problem. The paper is very well written with a clear description of the method and a sound motivation.
- The main contribution of the paper is using Matheron's rule to reduce the dependence of the number of tasks from a multiplicative factor to an additive one, leading to a large speedup. This is also verified experimentally showing at least an order of magnitude speedup without any approximations. Although Matheron's rule has been utilized in prior work, it has not been used for optimization of high-dimensional output functions, to my knowledge. As such, this contribution seems novel.
- The proposed method has been applied to a variety of settings - constrained multi-objective BO, and composite black box functions. Other than the computational speedup, experimental results also indicate improved sample complexity when modeling multi-output GPs as MTGPs rather than standard batched GPs. Overall the experimental results seem strong and the conclusions are interesting.

**Weaknesses:**
- Some prior work has not been discussed:
    * Uhrenholt AK, Jensen BS. Efficient Bayesian optimization for target vector estimation. AISTATS 2019
    * Chowdhury SR, Gopalan A. No-regret algorithms for multi-task bayesian optimization. AISTATS 2021
    * Matsui K, Kusakawa S, Ando K, Kutsukake K, Ujihara T, Takeuchi I. Bayesian Active Learning for Structured Output Design. arXiv preprint 2019
    * Likelihood free inference is also a closely related topic. For instance, see: Gutmann MU, Corander J. Bayesian optimization for likelihood-free inference of simulator-based statistical models. JMLR 2016.
    * Multi-objective BO references can be expanded: see the differentiable EHVI paper.

Minor concerns:
- P4 L158: should k(v_a, v_a') be k(v_(i_a), v_(j_a)) instead for a=1 to k?
- L228-230: Are the tables wrongly referenced?

**Time Spent Reviewing:**

6

---

> ### Author Response · Authors · 2021-08-10
> **Thanks for the positive feedback**
>
> Thank you for the thoughtful review as well as the pointers to very interesting related work which we had not seen previously. We will be sure to cite these in the camera ready as they represent a set of interesting multi-task BO problems and references.
>
> Here is a bit more context on each paper and how it relates to our work:
> - [Uhrenhold and Jensen, ‘19]: Target vector estimation can be re-written as a composite BO objective (see [Astudillo & Frazier, ‘19]) over the several output components in the vector with the cheap objective as the 2-norm. In fact, the environmental problem we study in Section 5 is a generalization of this problem to two-dimensional outputs.
> - [Chowdhury and Gopalan, ‘21]: Thanks for the pointer to this work, which is closely related to ours. They establish theoretical bounds with respect to maximization of the Pareto front (the set of values of non-dominated points across all objectives) for UCB in a multi-task setting with a Lipschitz constraint on the “composite function” (in our terminology) / “scalarization”. There are two distinctions from our work: first, our approach can take into account the batch setting where we acquire several points at once, and second, our approach is more broadly general: it applies to any differentiable acquisition function (not just UCB) and we can maximize the volume of the Pareto front (as we do in our multi-objective BO experiments) rather than simply maximizing the expected value of a scalarization (as is done in their approach).
> - [Matsui et al, ‘19]: This is another specialized multi-task problem that seeks to minimize the expected loss. With Matheron’s rule for posterior sampling, we expect to be able to expand their method to the batch acquisition setting where the distribution of the expected loss is likely intractable, as well as to larger design sizes.
> - [Guttmann and Corander, ‘16] (along with other likelihood free inference techniques): Thanks for the pointer to such an interesting paper! We did not know that was an application of Bayesian optimization, and we believe that efficient multi-task Bayesian optimization could prove useful for likelihood-free inference settings, particularly if there are many related posterior distributions to be inferred at once, as seems to be the case in many physics problems [Cranmer et al, ‘20].
> - Multi-objective problems: Indeed, thanks for the pointer. The usage of the expected hypervolume improvement dates back to Emmerich’s thesis [Emmerich, ‘05] with its computation being described in [Emmerich et al, ‘11], while EHVI was optimized in the context of Bayesian optimization with either exact gradients [Yang et al, ‘19] or approximate gradients [Wada and Hino, ‘19]. We will be sure to cite these in the camera ready. Finally, we believe that MTGPs could also help improve Bayesian optimization in the many-objective setting like in the work of [Binois et al, ‘20].
>
> Minor concerns:
> - Yes, thanks for the notational pointer in l. L158.
> - Also, yes, the reference should be to Table 1 and not Table 2.
>
> New References:
>
> Binois, Mickaël, et al. "The Kalai-Smorodinsky solution for many-objective Bayesian optimization." J. Mach. Learn. Res. 21.150 (2020): 1-42.
>
> Kyle Cranmer, Johann Brehmer, and Gilles Louppe. “The frontier of simulation-based inference.” Proceedings of the National Academy of Science 117 (2020).
>
> Emmerich, Michael. "Single-and multi-objective evolutionary design optimization assisted by gaussian random field metamodels." dissertation, Universität Dortmund (2005).
>
> Emmerich, Michael TM, André H. Deutz, and Jan Willem Klinkenberg. "Hypervolume-based expected improvement: Monotonicity properties and exact computation." 2011 IEEE Congress of Evolutionary Computation (CEC). IEEE, 2011.
>
> Wada, Takashi, and Hideitsu Hino. "Bayesian optimization for multi-objective optimization and multi-point search." arXiv preprint arXiv:1905.02370 (2019).
>
> Yang, Kaifeng, et al. "Multi-objective Bayesian global optimization using expected hypervolume improvement gradient." Swarm and evolutionary computation 44 (2019): 945-956.

---

> > ### Comment · Reviewer_zeBf · 2021-08-30
> > **More literature on MOBO**
> >
> > Thanks for addressing my concerns. Multi-objective optimization has a large body of prior work, beyond hypervolume improvement. I encourage the authors to also include literature other than HVI:
> > - Predictive Entropy Search for Multi-objective Bayesian Optimization, ICML 2016
> > - Max-value entropy search for multi-objective bayesian optimization, NeurIPS 2019
> > - A flexible framework for multi-objective bayesian optimization using random scalarizations, UAI 2019
> > - Multi-objective Bayesian Optimization using Pareto-frontier Entropy, 2019
> > - Diversity-Guided Multi-Objective Bayesian Optimization With Batch Evaluations, NeurIPS 2020

---

> > > ### Author Response · Authors · 2021-08-31
> > > **Thanks**
> > >
> > > Thanks for the pointers to these interesting references.
> > >
> > > With additional work, both Paria et al, UAI, '19 ("A flexible framework...") and Lukovic et al, NeurIPS, '20 ("Diversity-Guided Multi-Objective...") can be extended to use MTGPs rather than batch GPs; we can make a note of these when describing future work in the camera ready.

---

### Official Review · Reviewer_MXiX · 2021-07-15

**Rating:** 8
**Confidence:** 3

**Summary:**

The authors devise a novel method for exact multi-task Gaussian process (GP) sampling that dramatically improves time costs, from multiplicative to additive in the combination of tasks and data points. Their method works by exploiting Kronecker structures in covariance matrices and then applying Matheron's identity. This allows them to perform Bayesian optimisation (BO) for tens of thousands of correlated outputs, which was hardly possible previously.

**Limitations And Societal Impact:**

The authors extensively discussed the limitations of their work in the supplementary materials (a summary of this would have been helpful in the main text). This discussion also includes a very reasonable paragraph on societal impacts.

**Main Review:**

The authors set out to solve the problem of multi-task BO with an extremely large number of tasks, i.e. correlated outputs. While this idea is a known and important problem, I am not aware of any work that has attempted to solve multi-task BO with this many tasks. The authors combine two known techniques, i.e. that of exploiting Kronecker structures in covariances and using Matheron's identity, and apply them to the domain of multi-task GPs, yielding impressive results for posterior sampling times. The authors adequately cited related work in Section 1 and 2, but I believe this could have been improved by having a separate section that summarises related approaches more concisely.

The derivations appear technically sound to me, with well-supported theoretical claims, in particular with regards to sampling time costs. The experimental results were extensive and impressive. The authors tested their method on a variety of problems that each showcased different strengths of their method, e.g. sampling efficiency, applicability for multi-objective and constrained BO problems, as well as extremely large output dimensions.

This paper was clearly-written and well-motivated. The results are important, in my opinion, as they allow practitioners to perform BO in settings where there are an extremely large number of output dimensions. I believe this allows practitioners to start thinking about using BO in such settings which were previously infeasible.

Minor comments:
- The "Single-Output" time cost for Matheron's rule in Table 1 seems to be missing a bracket, or some symbols.
- Although the color schemes used in the experimental section look good, I think some of the colors are hard to distinguish. E.g. the purple and magenta lines in Figure 1, the Random, qParego-Batch and qEHVI-Batch lines in Figure 3, as well as the Random and EI lines in Figure 5.
- You don't say what the shaded area represents, e.g. is it 1 or 2 standard deviations?

**Time Spent Reviewing:**

4

---

> ### Author Response · Authors · 2021-08-10
> **Thanks for the positive feedback**
>
> Thank you for your very supportive review! We appreciate it. We respond to your points below. We also note we have an extended discussion of related work in Appendix B of the supplementary material and will signpost to this better in the camera ready.
>
> - “Single output”: Yes, there is an extra parenthesis.
> - Color schemes: Thanks for the suggestions, we will revise to make them more distinct: making the purple darker in Figure 2, and separating out the darker colors in Figure 3 and Figure 5 better.
> - Shading: All plots, unless otherwise specified, show the mean and two standard deviations of the mean across repeated trials. This is mentioned throughout the experimental section (both Section 4 and Appendix D), but we will feature it more prominently at the top of Section 4.

---

### Official Review · Reviewer_FR3J · 2021-07-17

**Rating:** 6
**Confidence:** 4

**Summary:**

This paper presents an efficient sampling method for multi-output Gaussian processes and demonstrates the benefits of the proposed sampling method on Bayesian optimization with high dimensional outputs. The proposed sampling method is based on the Matheron’s rule, which was brought into the attention of the ML community by Wilson et al. (2020), and extends the formulation into the multi-output GP scenario.

**Limitations And Societal Impact:**

The authors have discussed the various limitations of the proposed method.

**Main Review:**

This paper applies the Matheron’s rule to multi-output GPs and derives an efficient sampling method for multi-output GP in the exact formulation. It demonstrates the benefits of efficient multi-output GP sampling in the context of BO with high dimensional outputs. The derivation is not surprising but is a solid piece of work that can benefit the GP and BO community.

The paper is well written and easy to follow, while containing lots of details. In experiments, the paper compared the proposed method with baselines on multi-objective BO, constrained BO and BO with a composite objective function.

As the main contribution of this paper is the efficient sampling algorithm, which contains several numerical approximations, it would be nice to compare the drawn samples to the samples drawn from the exact multi-output GP formulation.

The trick of using Eigen decomposition of Kronecker product covariance matrix has been exploited in KISS-GP (Wilson&Nickisch 2015). With the Eigen decomposition trick, one can draw efficient samples on finite locations, which is sufficient for standard acquisition functions such as EI and UCB. I wonder what is the main improvement of the proposed method compared to this simple solution.

In the proposed method, the coregionalization matrix (the covariance matrix for tasks) is estimated with maximum likelihood. Due to the large number of parameters in the coregionalization matrix for high dimensional output, it often suffers from overfitting. Previous multi-task GP modeling or multi-task BO approaches tend to perform Bayesian inference for the coregionalization matrix such as MCMC sampling or variational inference. It would be interesting to compare the proposed method with other multi-task BO approaches (although those approaches might not be as scalable).

Some missing reference in the multi-output GP literature:

- “Kernels for vector-valued functions: A review”, MA Alvarez, L Rosasco, ND Lawrence, Foundations and Trends in Machine Learning, 2012.
- “Efficient Modeling of Latent Information in Supervised Learning using Gaussian Processes”, Z Dai, MA Álvarez, ND Lawrence, NeuRIPS 2017.


**Time Spent Reviewing:**

2

---

> ### Author Response · Authors · 2021-08-10
> **Thanks for the supportive feedback**
>
> Thank you for your supportive review! We address your questions below and make several clarifications. We hope you can consider our response in your final assessment.
>
> ***Sampling Procedure***
>
> The efficient sampling algorithm is actually “exact” (up to floating point precision) if we use eigen-decompositions for all of the matrix solves and posterior sampling, which we do use for all of the experiments where the number of total data points seen is fewer than 800 (GPyTorch’s default threshold for switching over to approximate numerical methods such as Lanczos and conjugate gradients). Specifically, Matheron’s rule for posterior sampling samples from exactly the same distribution as the GP posterior distribution does, it just does so in a different way. Indeed, we verify that posterior samples are equivalent to distributional sampling in Appendix Figure A.2, where we show the mean and standard deviation of the samples, as well as the estimated standard deviation from these samples.
>
> ***Distinction from KISS-GP and Usage of Posterior Sampling***
>
> KISS-GP [Wilson & Nickisch, ‘15] exploits Kronecker structure for solely low-dimensional single task datasets by performing interpolation against a set of grid structured inducing points.
> However, they do not study posterior sampling from the GP, and it was not until [Pleiss, et al, ‘18] that efficient sampling procedures from the posterior of KISS-GP were developed.
> Both works, however, only consider the single-task setting, rather than the multi-task setting we study. We will cite and reference both works in the camera ready, as they also exploit Kronecker structure in the context of Gaussian processes.
>
> However, our approach is completely amenable to combination with theirs if we consider using a multi-task version of KISS-GP, which we sketch next, with the goal of producing constant time samples for multi-task KISS-GP. For instance, [Stanton et al, ‘21] recently developed constant time updates to the training data covariance matrix which can be applied to updating the joint training covariance, (e.g. for producing \tilde R in line 683 in the Appendix). Then, we can again exploit Matheron’s rule and Kronecker structure as we did in our work.
>
> We note a couple of distinctions that require the usage of posterior sampling for BO. First, we consider batch acquisition functions (e.g. choosing several candidate points at once) that do not admit a closed-form expression and thus require joint posterior sampling. Further, we additionally require gradients of these samples with respect to the data as we use gradient based optimization to maximize the acquisition function with respect to the data.
> For further reading, please see [Wilson et al, ‘17] and [Balandat et al, ‘20].
>
> As a practical case, in the coverage map problem with the HOGP, we need to sample from a joint posterior over 5000 tasks because the objective function that we wish to maximize is actually a nonlinear function of the observed responses, so to compute the expected improvement the posterior over the objective is no longer a Gaussian distribution (but rather a nonlinear transformation of one) and we have to compute acquisition functions such as the expected improvement using Monte Carlo sampling, even if expected improvement would have been closed form for a single task GP.
>
> ***Co-regionalization Matrix and other Approaches***
>
> Overfitting is indeed a potential concern. If you can provide some pointers, we are happy to include any references where the inter-task covariance matrix is estimated using Bayesian methods. In order to avoid overfitting, we use LKJ priors (two informative explanations on the Internet are https://distribution-explorer.github.io/multivariate_continuous/lkj.html and https://docs.pymc.io/notebooks/LKJ.html with the original reference as [Lewandowski et al, ‘09]) to regularize the inter-task covariance matrix. These priors are popular in the Bayesian statistics community to estimate covariance matrices due to their flexibility and ease of understanding. The LKJ prior has a single hyper-parameter $\eta$ that controls the strength of the correlation between tasks, and we chose it to be $\eta = 2.0$ (which means that we regularize towards a diagonal task covariance matrix; $\eta = 1.0$ would correspond to an uninformative prior).
> In our experience, we’ve found MCMC estimation of both the task parameters and the kernel hyper-parameters to be too slow to be practical in comparison with the much quicker empirical Bayesian (optimization based) approach.
>
> Further, we do compare to other implementations of MT-BO in Figure 2: Our “Hadamard” implementation is exactly the same model (and implementation) as the LCE-M model of [Feng et al, ‘20], and, as suspected, scalability is much more limited although performance is similar, see Figures A.4 and A.5. Using this alternative model is computationally feasible for <= 10 tasks at once.
>
>
> ***References***
>
> Thank you for the pointers. We already cite [Alvarez et al, ‘19] in Appendix B.3 and will move that discussion to lines 131-149 in the main text. We will additionally cite [Dai et al, ‘17] in the camera ready; however, from our reading of that work, posterior sampling (using the posterior in Section 3.1 of that paper) will be very expensive (likely (nt)^3 due to the several summations in the posterior covariance preventing exploitation of Kronecker structure).
> As a note on the generality of our approach, we hypothesize that using Matheron’s rule conditioned on the inducing points (like the decoupled sampling approach of [Wilson et al, ‘20]) may improve the sampling speed.
>
> Lewandowski, Daniel, Dorota Kurowicka, and Harry Joe. "Generating random correlation matrices based on vines and extended onion method." Journal of multivariate analysis 100.9 (2009): 1989-2001.
>
> Pleiss, Geoff, et al. "Constant-time predictive distributions for Gaussian processes." International Conference on Machine Learning. PMLR, 2018.
>
> Stanton, Samuel, et al. "Kernel Interpolation for Scalable Online Gaussian Processes." International Conference on Artificial Intelligence and Statistics. PMLR, 2021.
>
> Wilson, Andrew, and Hannes Nickisch. "Kernel interpolation for scalable structured Gaussian processes (KISS-GP)." International Conference on Machine Learning. PMLR, 2015.

---

> > ### Author Response · Authors · 2021-08-24
> > **Checking in**
> >
> > Hi reviewer FR3J,
> >
> > We would be grateful if you could confirm whether our response has addressed your concerns, and let us know if any issues remain.To recap, we
> > - clarified that our sampling algorithm is indeed exact
> > - discussed how our approach relates to KISS-GP
> > - described how we use an LKJ prior on the inter-task covariance to avoid overfitting
> > - will move of the “missing” references to the main text
> >
> > Thanks,
> >
> > The Authors

---

### Author Response · Authors · 2021-08-10
**To All Reviewers**

**To All Reviewers**

We thank all of the reviewers for their feedback, and would like to emphasize a couple of points.

In our paper, we propose a new sampling method for multi-task Gaussian processes (MTGPs) that scales additively in the combination of tasks and data points by exploiting Matheron’s rule for sampling conditional Gaussian distributions. We’ve validated the implementation by extensive experiments, as well as demonstrating the broad applicability and success of MTGPs to a range of Bayesian optimization tasks. All reviewers agreed that the approach was novel, timely, and performed well across the suite of tasks.

While MTGPs have been reasonably well studied in the ML community for the past decade, we believe that the expensive computational scaling (the cubic, multiplicative scaling in the number of data points and tasks) has significantly limited their adoption. By using Matheron’s rule sampling, we are able to generalize multi-task Bayesian optimization to thousands of tasks, well beyond what was previously feasible. This allows practitioners to begin thinking about Bayesian optimization of functions with high-dimensional output as a problem that is now solvable. We therefore believe the contribution has particularly large practical significance.

Furthermore, as we sketched in Appendix C.1.1 and demonstrated in our reply to Reviewer Dpyw, the computational improvements of Matheron’s rule for posterior sampling in MTGPs are not just limited to the intrinsic model of coregionalization (ICM) kernels. Using Matheron’s rule for posterior sampling the linear model of co-regionalization (LMC) also provides significant computational gains against distributional sampling.

We respond to each reviewer in separate posts. We hope our response, and the practical significance of the work, can be taken into account in the final assessment.

---

### Decision · Program_Chairs · 2021-09-27

**Decision:**

Accept (Poster)

**Comment:**

The reviewers have provided thoughtful and constructive comments. They have responded to the authors' feedback and have reached consensus to recommend acceptance. I hope the authors will take the reviewers' comments to heart and encourage them to incorporate their thoughts in preparing the camera-ready version of their manuscript.